# Chromatin-accessibility estimation from single-cell ATAC-seq data with scOpen

Zhijian Li [1,7], Christoph Kuppe [2,3,7], Susanne Ziegler[2], Mingbo Cheng[1], Nazanin Kabgani[2], Sylvia Menzel[2], Martin Zenke[4,5], Rafael Kramann [2,3,6 ✉] & Ivan G. Costa [1 ✉]

A major drawback of single-cell ATAC-seq (scATAC-seq) is its sparsity, i.e., open chromatin regions with no reads due to loss of DNA material during the scATAC-seq protocol. Here, we propose scOpen, a computational method based on regularized non-negative matrix factorization for imputing and quantifying the open chromatin status of regulatory regions from sparse scATAC-seq experiments. We show that scOpen improves crucial downstream analysis steps of scATAC-seq data as clustering, visualization, *cis*-regulatory DNA interactions, and delineation of regulatory features. We demonstrate the power of scOpen to dissect regulatory changes in the development of fibrosis in the kidney. This identifies a role of Runx1 and target genes by promoting fibroblast to myofibroblast differentiation driving kidney fibrosis.

---

[1] Institute for Computational Genomics, Joint Research Center for Computational Biomedicine, RWTH Aachen University Medical School, 52074 Aachen, Germany. [2] Institute of Experimental Medicine and Systems Biology, RWTH Aachen University Medical School, 52074 Aachen, Germany. [3] Division of Nephrology and Clinical Immunology, RWTH Aachen University, 52074 Aachen, Germany. [4] Department of Cell Biology, Institute of Biomedical Engineering, RWTH Aachen University Medical School, 52074 Aachen, Germany. [5] Helmholtz Institute for Biomedical Engineering, RWTH Aachen University, Aachen, Germany. [6] Department of Internal Medicine, Nephrology and Transplantation, Erasmus Medical Center, 3015GD Rotterdam, The Netherlands. [7] These authors contributed equally: Zhijian Li, Christoph Kuppe. ✉email: rkramann@gmx.net; ivan.costa@rwth-aachen.de

The simplicity and low cell number requirements of the assay for transposase-accessible chromatin using sequencing (ATAC-seq)[1] made it the standard method for detection of open chromatin (OC), enabling the first study of OC of cancer cohorts[2]. Moreover, careful consideration of digestion events by the enzyme Tn5 allowed insights on regulatory elements such as positions of nucleosomes[1,3], transcription factor (TF) binding sites, and the activity level of TFs[4]. The combination of ATAC-seq with single-cell sequencing (scATAC-seq)[5] further expanded ATAC-seq applications by measuring the OC status of thousands of single cells from healthy[6,7] and diseased tissues[8]. Computational tasks for analysis of scATAC-seq include detection of cell types with clustering (scABC[9], cisTopic[10], SnapATAC[11]); identification of TF regulating individual cells (chromVAR[12]); and prediction of co-accessible DNA regions in groups of cells (Cicero[13]).

Usually, the first step for analysis of scATAC-seq data is the detection of OC regions by calling peaks on the scATAC-seq library by ignoring cell information. Next, a matrix is built by counting the number of digestion events per cell in each of the previously detected regions. This matrix usually has a very high dimension (up to >$10^6$ regions) and a maximum of two digestion events are expected for a region per cell. As with scRNA-seq[14–16], scATAC-seq is affected by dropout events due to the loss of DNA material during library preparation. These characteristics render the scATAC-seq count matrix sparse, i.e. 3% of non-zero entries. In contrast, scRNA-seq have less severe sparsity (>10% of non-zeros) than scATAC-seq due to smaller dimension (< 20,000 genes for mammalian genomes) and lower dropout rates for genes with high or moderate expression levels. This sparsity poses challenges in the identification of cell-specific OC regions and is likely to affect downstream analysis as clustering and detection of regulatory features. Although several computational methods have been developed to address this issue for scRNA-seq data (e.g., MAGIC[14], scImpute[17], DCA[18], and SAVER[19]), these methods were not designed to deal with the sparse and low count nature of scATAC-seq data. Until date, there are only two approaches for imputation methods for scATAC-seq data e.g., SCALE[20] and cisTopic[10]. SCALE, which is based on deep learning, requires a graphics processing unit (GPU) for training. The usual small size of GPU memory limits the number of cells to be analyzed. cisTopic is a Bayesian-based method, which was reported to have an exponential increase of the running time for an increasing number of reads[21]. Therefore, both approaches are likely to have scalability issues with large data sets.

We here present scOpen, an unsupervised learning model for scATAC-seq data imputation. It estimates accessibility scores to indicate if a region is open in a particular cell. scOpen is based on a non-negative matrix factorization (NMF), which makes no assumption on the data distribution as SCALE or cisTopic. It also includes a regularization, which makes it less prone to overfitting. To speed up the learning, we make use of a cyclic coordinate descent (CCD) algorithm. Moreover, we adopt an elbow detection approach to automatically determine the number of dimensions of the input data. The imputed matrix can be used as input for usual computational methods of scATAC-seq data as clustering, visualization, and prediction of DNA interactions (Fig. 1a). We demonstrate the power of scOpen on a comprehensive benchmarking analysis using publicly available scATAC-seq data with true labels. Moreover, we use scOpen together with HINT-ATAC[4] footprinting analysis to infer regulatory networks driving the development of fibrosis with a scATAC-seq time-course dataset of 31,000 cells in murine kidney fibrosis, identifying Runx1 as a regulator of myofibroblast differentiation.

## Results

**OC estimation with scOpen.** scOpen performs imputation and denoising of a scATAC-seq matrix via a regularized NMF based on a binarized scATAC-seq cell count matrix, where features represent OC regions which are obtained by peak calling based on aggregated scATAC-seq profiles. This matrix is transformed using the term frequency-inverse document frequency (TF-IDF), which weighs the importance of an OC region to a cell. Next, it applies a regularized NMF using a coordinate descent algorithm[22]. In addition, it provides a computational approach to optimize the dataset-specific rank $k$ of the NMF approach based on a knee detection method[23]. scOpen provides as results imputed and reduced dimension matrices, which can be used for distinct downstream analysis as visualization, clustering, inference of regulatory players, and *cis*-regulatory DNA interactions (Fig. 1a).

First, we made use of simulated scATAC-seq similar as in ref. [21] to evaluate the parameterization of two hyper-parameters of scOpen, i.e., the rank $k$ and the regularization term $\lambda$ (see the "Methods" section; Supplementary Fig. 1a–d). Results indicate that the scOpen automatic procedure for rank selection obtains close to optimal results, i.e. selected rank had similar accuracy than best ranks for both imputation and clustering problems. Regarding $\lambda$, a value of 1 is optimal in the imputation problem, where values in the range [0, 1] were optimal for the clustering problem. This indicates the importance of the regularization parameter in scATAC-seq data imputation. The $\lambda = 1$ and the rank selection strategy are used as default by scOpen.

**Benchmarking of scOpen for imputation of scATAC-seq.** For benchmarking, we made use of four public scATAC-seq data sets: cell lines[5], human hematopoiesis composing of eight cell types[6], four sub-types of T cells[8], and a multi-omics RNA-ATAC from peripheral blood mononuclear cells (PBMCs) with 14 cell types (see the "Methods" section). These datasets were selected due to the presence of external labels, which were defined independently of the scATAC-seq at hand. After processing, we generated a count matrix for each dataset and detected 50k to 120k OC regions with 3–7% of non-zero entries, confirming the sparsity of scATAC-seq data (Supplementary Table 1). For comparison, we selected top-performing imputation/denoising methods[24] proposed for scRNA-seq (MAGIC[14], SAVER[19], scImpute[17], DCA[18], and scBFA[25]); two scATAC-seq imputation methods (cisTopic[10] and SCALE[20]); a PCA-based imputation method (imputePCA[26]); and the raw count matrix (Supplementary Fig. 2a).

We first evaluated the time and memory requirement of imputation methods (see the "Methods" section). scOpen had the overall lowest memory requirements, i.e it required at least 2 fold less memory as compared to cisTopic, MAGIC, or SCALE (Fig. 1b) and had a maximum requirement of 16 GB on the PBMC dataset (Supplementary Data 1). Regarding computing time, MAGIC was the fastest followed by SCALE and scOpen. These were the only methods performing the imputation of the large PBMCs dataset (10k cells vs. 100k peaks) in < 3 h (Fig. 1c), while imputePCA, SAVER, and DCA failed to execute at the PBMCs dataset.

We next tested if imputation methods can improve the recovery of true OC regions. For this, we created true and negative OC labels for each cell type by peak calling of bulk ATAC-seq profiles. Next, we evaluated the correspondence between imputed scATAC-seq values and peaks of the corresponding cell type with the area under precision-recall curve (AUPR) (see the "Methods" section). scOpen significantly outperformed all competing methods by presenting the highest mean AUPR (Fig. 1d). The combined ranking indicates SCALE

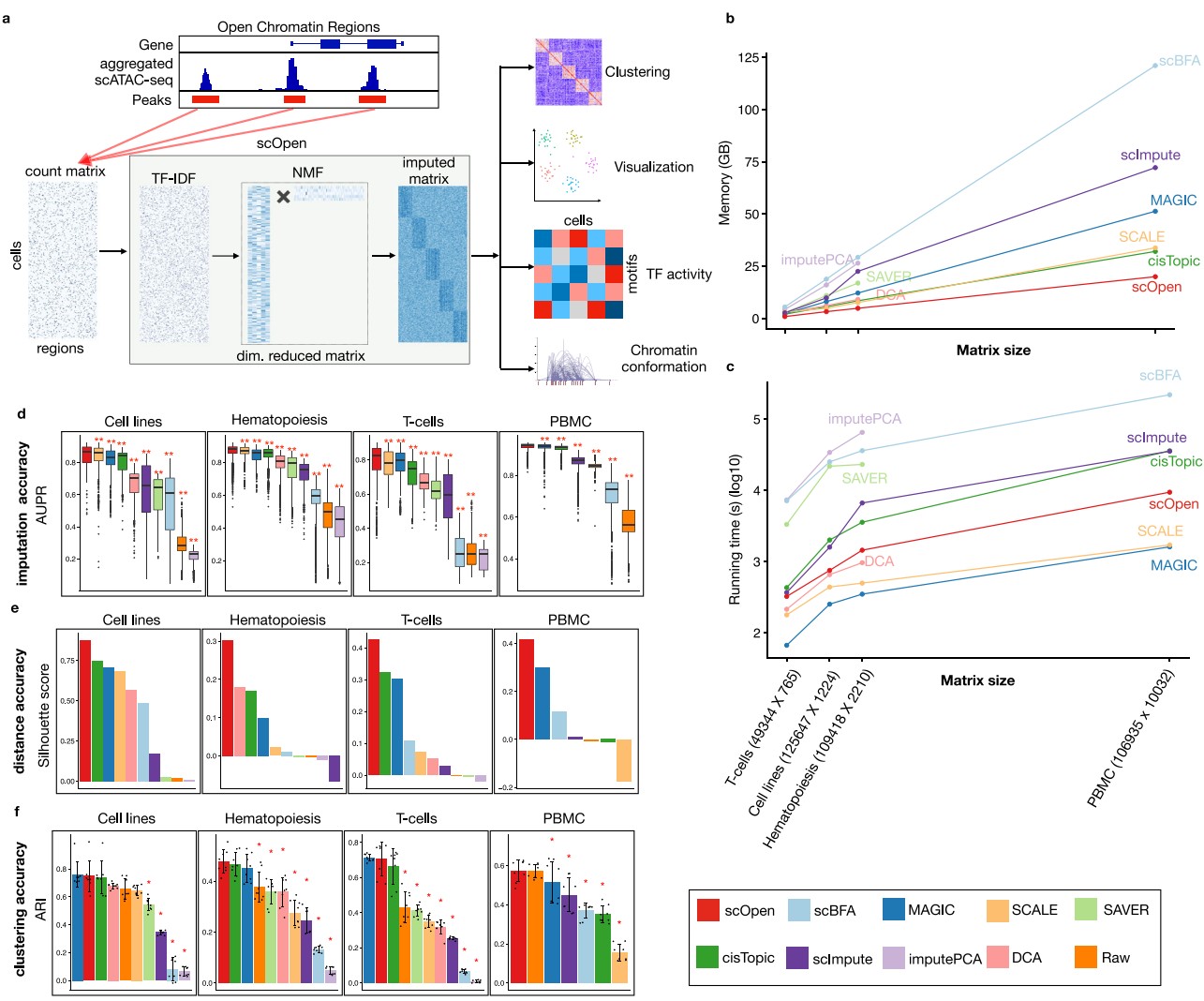

**Fig. 1 scOpen and benchmarking of scATAC-seq imputation methods. a** scOpen receives as input a sparse peak by cell count matrix. After matrix binarization, scOpen performs TF–IDF transformation followed by NMF for dimension reduction and matrix imputation. The imputed or reduced matrix can then be given as input for scATAC-seq methods for clustering, visualization, and interpretation of regulatory features. **b** Memory requirements of imputation/denoising methods on benchmarking datasets. The x-axis represents the number of elements of the input matrix (number of OC regions by cells). **c** Same as **b** for running time requirements. **d** Boxplot showing the evaluation of imputation/denoising methods for recovering true peaks. The y-axis indicates the area under the precision-recall curve (AUPR). Methods are ranked by the mean AUPR. The asterisk and the two asterisks mean that the method is outperformed by the top-ranked method (scOpen) with significance levels of 0.05 and 0.01 at a confidence level of 0.95 (Wilcoxon Rank Sum test, paired, two-sided), respectively ($n = 1224$ cells for Cell lines, $n = 2210$ cells for Hematopoiesis, $n = 765$ cells for T-cells, and $n = 10,032$ for PBMC). The box plot represents the median (central line), first and third quartiles (box bounds). The whiskers present the 1.5 interquartile range (IQR) and external dots represent outliers (data greater than or smaller than 1.5IQR). **e** Barplots showing silhouette score (y-axis) for benchmarking datasets. **f** Barplots showing the clustering accuracy for distinct imputation methods. The y-axis indicates the mean adjusted Rand Index (ARI). Dots represent individual ARI values of distinct clustering methods. Error bars represent the standard deviation (SD) of ARI. Data are represented as mean ± SD. The asterisk and the two asterisks mean that the method is outperformed by the top-ranked method with significance levels of 0.05 and 0.01 at a confidence level of 0.95 ($n = 8$ independent clustering experiments, Wilcoxon Rank Sum test, paired, two-sided), respectively. Source data for Fig. 1 are provided as a Source Data file.

and MAGIC as runner-up methods (Supplementary Fig. 2b). Next, we evaluated the influence on the number of cells per cluster in the AUPR. Despite an overall decrease in AUPR with sample size, we observed that top performing methods (scOpen, SCALE, and MAGIC) were less sensible to cell numbers (Supplementary Fig. 2c).

We also investigated the impact of imputation on the estimation of distances between cells and the impact on standard clustering methods. Distance between cells was evaluated with the silhouette score, while clustering accuracy was evaluated with adjusted Rand index (ARI)[27] both regarding the agreement with

known cell labels. scOpen was the best performer in all data sets regarding the silhouette score (Fig. 1e). The combined ranking demonstrated that scOpen had significantly better results than competing methods, while cisTopic and MAGIC were runner-up methods (Supplementary Fig. 2d). Regarding clustering, scOpen was best in the hematopoiesis and multi-omics PBMCs datasets and second-best for cell lines and T cell datasets (Fig. 1f). When considering the combined ranking, scOpen performed best followed by cisTopic and MAGIC (Supplementary Fig. 2e). Visual representations with UMAP[28] projections of these datasets and methods are provided in Supplementary Fig. 3. Altogether,

these results support that scOpen outperforms state-of-the-art imputation methods while providing the lowest memory footprint and above-average time performance.

**Benchmarking of scATAC-seq clustering methods**. Another relevant question was to compare scOpen with top-performing state-of-the-art scATAC-seq pipelines: cisTopic, SnapATAC and Cusanovich2018[21] (see the "Methods" section; Supplementary Fig. 4a). Here, pipelines were evaluated with the default clustering methods, i.e graph-based clustering for SnapATAC[11] and density-based clustering for other methods[10]. We also evaluated the use of both reduced and imputed matrices for scOpen and cisTopic, as these methods provide both types of representations.

The evaluation of distance matrices with the silhouette score indicated that both imputed or low dimension scOpen matrices presented the highest score in all data-sets (Fig. 2a) and both scOpen matrix representations tied as first in the combined rank (Supplementary Fig. 4b). cisTopic, which was the runner-up method, performed well in cell lines, hematopoiesis, and T-cells but poorly for multi-omics PBMCs. Next, we evaluated the clustering performance of competing pipelines. Again, scOpen performed best on cell lines and hematopoiesis data sets and ranked first/second in the combined rank (Supplementary Fig. 4c). Overall, this analysis indicates that both reduced dimension and imputed scOpen matrices obtain the best overall results for distance and clustering representations on evaluated datasets. Of note, the low-dimensional matrix reduces the memory footprint on clustering by more than 1000 fold in comparison to using full imputed matrices and serves as an alternative for cluster analysis of large dimensional data sets.

**Improving scATAC-seq downstream analysis using scOpen estimated matrix**. Next, we tested the benefit of using scOpen estimated matrices as input for scATAC-seq computational pipelines, which have as objective the identification of regulatory features associated with single cells (chromVAR[12]), estimation of gene activity scores and DNA-interactions (Cicero[13]), or a clustering method tailored for scATAC-seq data (scABC[9]) (Supplementary Fig. 4d). Both chromVAR and Cicero first transform the scATAC-seq matrix to either TFs and genes feature spaces respectively. Clustering was then performed using the standard pipelines from each approach. We compared the clustering accuracy (ARI) and distance (silhouette score) of these methods with either raw or scOpen estimated matrices. In all combinations of methods and datasets, we observed a higher or equal ARI/silhouette whenever a scOpen matrix was provided as input (Fig. 2c, d). Results can be inspected with UMAP visualization with and without scOpen imputation (Supplementary Fig. 5).

Prior to estimating gene-centric OC scores, Cicero first predicts co-accessible pairs of DNA regions in groups of cells, which potentially form *cis*-regulatory interactions. We compared Cicero predicted interactions on human lymphoblastoid cells (GM12878) by using Hi-C and ChIA-PET from this cell type as true labels for all imputation methods with data as provided in ref. [13]. Both AUPR values and odds ratios indicated that the scOpen matrix improves the detection of GM12878 interactions globally (Fig. 2e, f; Supplementary Fig. 6a, b). To evaluate the impact on the number of cell on these predictions, we have down-sampled the data to only consider 50% or 25% of cells. We observed a residual decrease in the AUPR of scOpen for 25% of cells (Supplementary Fig. 6c). This supports that chromatin conformation prediction works well even for cell types with low abundance. The power of scOpen imputation was clear when checking the individual locus (Fig. 2g), as previously described by Cicero[13]. This is evident when

contrasting accessibility scores between pairs of peak-to-peak links supported by Hi-C predictions (Fig. 2h; Supplementary Fig. 6d–h). scOpen obtained highly correlated accessibility scores, while other imputation methods showed quite diverse association patterns. Together, these results indicated that the use of scOpen estimated matrices improves downstream analysis of state-of-the-art scATAC-seq methods.

**Applying scOpen to scATAC-seq of fibrosis driving cells**. Next, we evaluated scOpen in its power to improve the detection of cells in a complex disease dataset. For this, we performed whole mouse kidney scATAC-seq in C57Bl6/WT mice in homeostasis (day 0) and at two-time points after injury with fibrosis: 2 and 10 days after unilateral ureteral obstruction (UUO)[29,30]. Experiments recovered a total of 30,129 high-quality cells after quality control with an average of 13,933 fragments per cell, a fraction of reads in promoters of 0.46, and high reproducibility ($R > 0.99$) between biological duplicates (Supplementary Fig. 7a, b; Supplementary Table 1). After data aggregation, 150,593 peaks were detected, resulting in a highly dimensional and sparse scATAC-seq matrix (4.2% of non-zeros).

Next, we performed data integration for batch effect removal using Harmony[31]. For comparison, we used a dimension reduced matrix from either LSI (Cusanovich2018), cisTopic, SnapATAC, or scOpen. We annotated the scATAC-seq profiles using single nuclei RNA-seq (snRNA-seq) data of the same kidney fibrosis model from an independent study[32] via label transfer[33] to serve as cell labels. We then evaluated the batch correction results using silhouette score and clustering. We observed that clusters based on scOpen were more similar to the transferred labels (higher ARI) than clusters based on competing methods (Fig. 3a). Furthermore, scOpen also provided better distance metrics and visualization than competing methods (Supplementary Figs. 7c–e and 8). These results support the discriminative power of scOpen in this large and complex dataset.

Next, we annotated the clusters of scOpen by using known marker genes and transferred labels after removing doublets with ArchR[34]. We identified all major kidney cell types including PT cells, distal/connecting tubular cells, collecting duct and loop of Henle, endothelial cells (ECs), fibroblasts as well as the rare populations of podocytes and lymphocytes (Fig. 3b; Supplementary Fig. 9a). Lymphocytes were not described in the previously scRNA-seq study[32], which supports the importance of annotation of scATAC-seq clusters independently of scRNA-seq label transfer. Of particular interest were cell types with population changes during the progression of fibrosis (Fig. 3c; Supplementary Fig. 9b–d). We observed an overall decrease of normal proximal tubular (PT), glomerular and ECs and an increase of immune cells as expected in this fibrosis model with tubule injury, the influx of inflammatory cells, and capillary loss[35,36]. Importantly, we detected an increased PT sub-population, which we characterized as injured PT by increased accessibility around the PT injury markers *Vcam1* and *Kim1* (*Havrc1*)[37] (Supplementary Fig. 9a).

**Dissecting cell-specific regulatory changes in fibrosis**. Next, we adapted HINT-ATAC[4] to dissect regulatory changes in scATAC-seq clusters. For each cluster, we created a pseudo-bulk ATAC-seq library by combining reads from single cells in the cluster. We then performed footprinting analysis and estimated TF activity scores for all footprint-supported motifs. We only kept TFs with changes (high variance) in TF activity scores among clusters. We focused here on clusters associated with PT cells, fibroblasts, and immune cells, as these represent key players in kidney remodeling

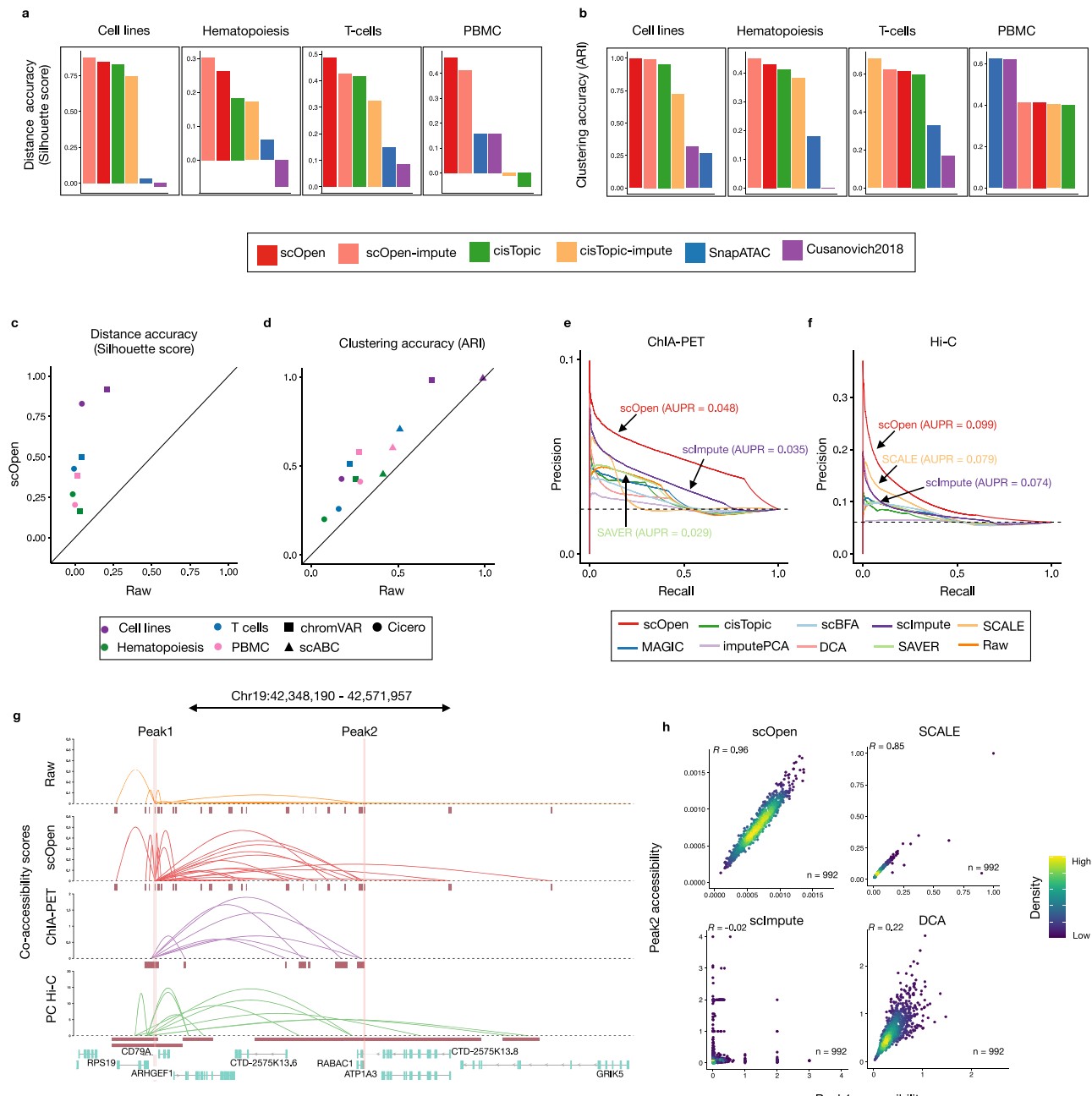

**Fig. 2 Benchmarking of scATAC-seq clustering and downstream analysis. a** Bar plot showing an evaluation of distances estimated on distinct scATAC-seq representations with a silhouette score. **b** Bar plots showing the clustering accuracy (ARI) for distinct clustering pipelines. **c** Scatter plot comparing silhouette score of datasets by providing raw (x-axis) and scOpen estimated matrices (y-axis) as input for Cicero and chromVAR. Colors represent datasets and shapes represent methods. scABC is not evaluated as it does not provide a space transformation. **d** Same as **c** for clustering results (ARI) of Cicero, chromVAR, and scABC. **e** Precision-recall curves showing the evaluation of the predicted links on GM12878 cells using the raw and imputed matrix as input. We used data from pol-II ChIA-PET as true labels. Colors refer to methods. We reported the AUPR for the top 3 methods. **f** Same as **e** by using Hi-C data as true labels. **g** Visualization of co-accessibility scores (y-axis) of Cicero predicted with raw and scOpen estimated matrices contrasted with scores based on RNA pol-II ChIA-PET (purple) and promoter capture Hi-C (green) around the CD79A locus (x-axis). For ChIA-PET, the log-transformed frequencies of each interaction PET cluster represent co-accessibility scores, while the negative log-transformed p-values from the CHiCAGO software indicate Hi-C scores. **h** Scatter plot showing single-cell accessibility scores estimated by top-performing imputation methods (according to **f**) for the link between peak 1 and peak 2 (supported by Hi-C data). Each dot represents a cell and color refers to density. Pearson correlation is shown on the left-upper corner. Source data for Fig. 2 are provided as a Source Data file.

and fibrosis after injury. As shown in Fig. 3d, the TF activity scores capture regulatory programs associated with these three major cell populations (Supplementary Data 2). Injured PTs have overall lower TF activity scores than all TFs of the PT cluster. TFs with a high decrease in activity in injured PTs include Rxra, which is

important for the regulation of calcium homeostasis in tubular cells[38], and Hnf4a, which is important in PT development[39] (Fig. 3d, e). Footprint profile of Rxra in injured PTs display a gradual loss of TF activity over time indicating that injured PT acquires a de-differentiated phenotype during fibrosis progression

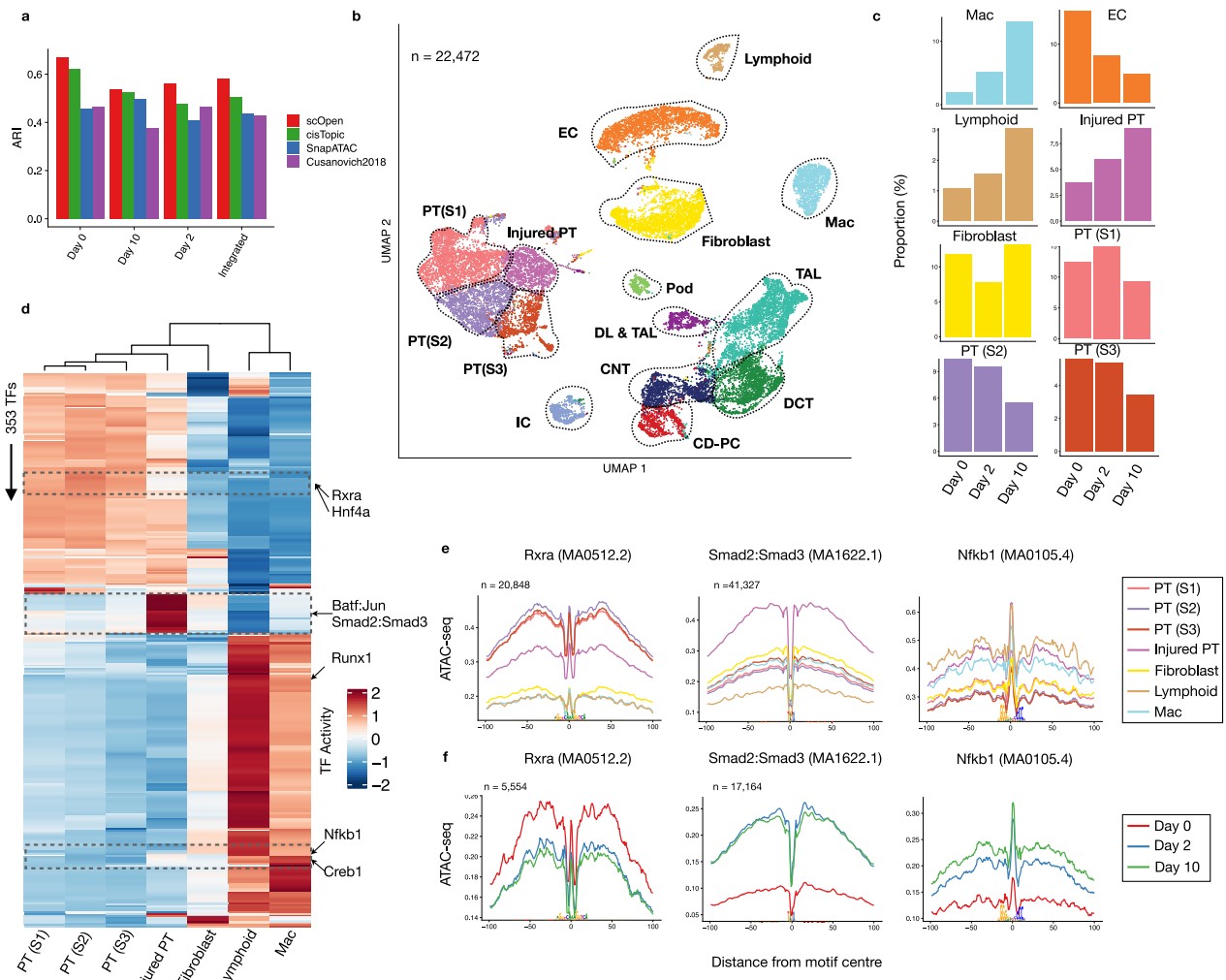

**Fig. 3 scOpen characterizes the progression of kidney fibrosis. a** ARI values (*y*-axis) contrasting clustering results and transferred labels using distinct dimensional reduction methods for scATAC-seq. Clustering was performed by only considering UUO kidney cells on day 0 (WT), day 2, or day 10 or the integrated data set (all days). **b** UMAP of the integrated UUO scATAC-seq after doublet removal with major kidney cell types: fibroblasts, descending loop of Henle and thin ascending loop of Henle (DL & TAL); macrophages (Mac), Lymphoid (T and B cells), endothelial cells (EC), thick ascending loop of Henle (TAL), distal convoluted tubule (DCT), collecting duct-principal cells (CD-PC), intercalated cells (IC), podocytes (Pod) and proximal tubule cells (PT S1; PT S2; PT S3; Injured PT). **c** Proportion of cells of selected clusters on either day 0, day 2 or day 10 experiments. **d** Heatmap with TF activity score (*z*-transformed) for TFs (*y*-axis) and selected clusters (*x*-axis). We highlight TFs with the decrease in activity scores in injured PTs (Rxra and Hnf4a), with high TF activity scores in injured PTs (Batf:Jun; Smad2:Smad3) and immune cells (Creb1; Nfkb1). **e** Transcription factor footprints (average ATAC-seq around predicted binding sites) of Rxra, Smad2::Smad3 and Nfkb1 for selected cell types. The logo of underlying sequences is shown below and the number of binding sites is shown top-left corner. **f** Transcription factor footprints of Rxra, Smad2::Smad3, and Nfkb1 for injured PT cells in day 0, day 2, and day 10. Source data for Fig. 3 are provided as a Source Data file.

and tubular dilatation (Fig. 3f). A group of TFs with high activity scores in injured PTs also have increased TF activity scores in fibroblasts (Smad2:Smad3 and Batf:Jun) indicating shared regulatory programs in these cells. Smad proteins are downstream mediators of TGFβ signaling, which is a known key player of fibroblast to myofibroblast differentiation and fibrosis[40]. The high activity of Smad2:Smad3 also indicates a role of TGFβ in the de-differentiation of injured PTs. Also, both Smad2:Smad3 reach a peak in TF activity level at day 2 after UUO in injured PTs (Fig. 3f), which indicates these TFs are activated post-transcriptionally. We also detect the high activity of Nfkb1 in injured PTs (and lymphocytes), which fits with the known role of Nfkb1 in injured and failed repair PTs[41,42]. Moreover, our analysis also shows a gradual TF activity increase over time in injured PT (Fig. 3f), suggesting that Nfkb1 plays an important role in sustaining the injured PT phenotype.

**scOpen reveals TF driving myofibroblast differentiation.** A key process in kidney injury is fibrosis, which is caused by the differentiation of fibroblasts and pericytes to matrix secreting myofibroblasts[43]. To dissect potential differentiation trajectories, we performed a diffusion map embedding of the fibroblasts (Fig. 4a), which revealed the presence of three major branches formed by fibroblasts, pericytes, and myofibroblasts, as supported by the accessibility of *Scara5*, *Ng2* (*Cspg4*), *Postn* and *Col1a1* (Supplementary Fig. 10)[43,44].

We next created a cellular trajectory across the differentiation from fibroblasts to myofibroblasts using ArchR (Fig. 4a; Supplementary Fig. 10c). We observed that there is an increase in cells after injury (Day 2 and Day 10) along the trajectory (Fig. 4b). We next characterized TFs by correlating their gene activity with TF activity along the trajectory (Fig. 4c) and ranked these by their correlation (Supplementary Fig. 10d). The

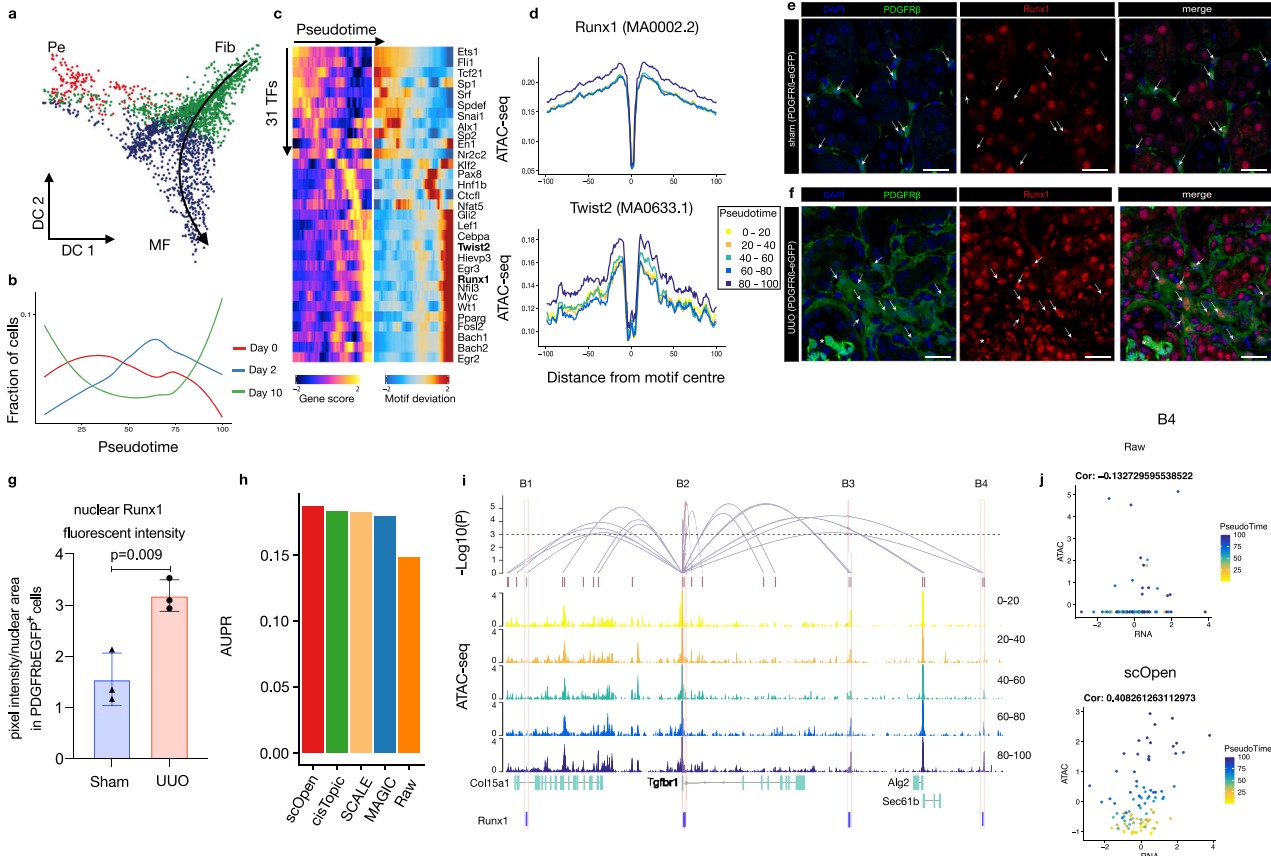

**Fig. 4 Role of Runx1 in myofibroblast differentiation. a** Diffusion map showing sub-clustering of fibroblasts. Colors refer to sub-cell-types and arrow represents differentiation trajectory from fibroblast to myofibroblast. Pe pericyte, Fib fibroblast, MF myofibroblast. **b** Line plots showing cell proportion from the day after UUO along the trajectory. **c** Pseudotime heatmap showing gene activity (left) and TF motif activity (right) along the trajectory. **d** Footprinting profiles of Runx1 and Twist2 binding sites along the trajectory. **e** Immuno-fluorescence (IF) staining of Runx1 (red) in PDGFRb-eGFP mouse kidney. In sham-operated mice, Runx1 staining shows a reduced intensity in PDGFRb-eGFP+ cells compared to remaining kidney cells (arrows). **f** Immuno-fluorescence (IF) staining of Runx1 (red) in PDGFRb-eGFP mouse kidney at 10 days after UUO as compared to sham. Arrows indicate Runx1 staining in expanding PDGFRb-eGFP+ myofibroblasts. **g** Quantification of Runx1 nuclear intensity in PDGFRb-eGFP+ cells in sham vs. UUO mice. Error bars represent the SD of the intensity. Data are presented as mean ± SD. Statistical significance was assessed by a two-tailed Student's *t*-test with *p* < 0.05 being considered statistically significant (*n* = 3 mice). **h** Performance of top-performing imputation methods on the prediction of Runx1 target genes measured with AUPR. **i** Peak-to-Gene links (top) predicted on scOpen matrix and associated to Tgfbr1 in fibroblast cells. The height of links represents its significance. Dash line represents the threshold of significance (FDR = 0.001). ATAC-seq tracks (below) were generated from pseudo-bulk profiles of fibroblast/myofibroblast cells with increasing pseudo time (0–20, 20–40, 40–60, 60–80, and 80–100). Binding sites of Runx1 (B1–B4) supported by ATAC-seq footprints and overlapping to peaks are highlighted on the bottom. **j** Scatter plot showing gene activity of Tgfbr1 and normalized peak accessibility from raw (upper) or scOpen imputed matrix (lower) for peak-to-gene link B4. Each dot represents cells in a given pseudotime and the overall correlation is shown in the left-upper corner. Scale bars in **e** and **f** represent 50 μm. For details on statistics and reproducibility, see the "Methods" section. Source data for Fig. 4 are provided as a Source Data file.

correlation of Runx1, which has a well-known function in blood cells[45], stood out, besides showing a steady increase in activity in myofibroblasts. Another TF with high correlation and similar myofibroblast specific activity was Twist2, which has a known role in epithelial to mesenchymal transition in kidney fibrosis[46] (Fig. 4d).

To validate the yet uncharacteristic role of Runx1 in myofibroblasts, we performed immunostaining and quantification of Runx1 signal intensity in transgenic PDGFRb-eGFP mice that genetically tag fibroblasts and myofibroblasts[43,47]. Runx1 staining in control mice (sham) revealed positive nuclei in tubular epithelial cells and rarely in PDGFRb-eGFP+ mesenchymal cells (Fig. 4e). In kidney fibrosis after UUO surgery (day 10), Runx1 staining intensity increased significantly in PDGFRb+ myofibroblasts (Fig. 4f, g). Next, we performed retroviral overexpression experiments and RNA-sequencing in a human kidney PDGFRb+

fibroblast cell-line that we have generated[43] to ask whether Runx1 might be functionally involved in myofibroblast differentiation in humans (Supplementary Fig. 11a, b). Runx1 overexpression led to reduced proliferation (Supplementary Fig. 11c) and strong gene expression changes (Supplementary Fig. 11d). Gene ontology (GO) and pathway enrichment analysis indicated enrichment of cell adhesion, cell differentiation, and TGFB signaling following Runx1 overexpression (Supplementary Fig. 11e). Various extracellular matrix genes (*Fn1*, *Col13A1*), as well as a TGFB receptor (*Tgfbr1*) and Twist2, were up-regulated following Runx1 overexpression (Supplementary Fig. 11d). Furthermore, we observed increased expression of the myofibroblast marker gene *Postn* after Runx1 overexpression. Altogether, this suggests that Runx1 might directly drive myofibroblast differentiation of human kidney fibroblasts since overexpression reduced cell proliferation and induced expression of various myofibroblast genes.

**Identification of Runx1 target genes**. Another important application of scATAC-seq is the prediction of *cis*-regulatory DNA interactions (peak-to-gene links) by measuring the correlation between gene activity and reads counts in proximal peaks. To compare the impact of imputation on this task, we predicted peak-to-gene links in fibroblasts on distinct scATAC-seq matrices using ArchR[34] after imputation with top-performing imputation methods. The use of imputation methods led to improved signals on peak-to-gene links predictions as indicated by higher correlation values after imputation (Supplementary Fig. 12a, b). We considered all genes with at least one link, where the peak has a footprint supported Runx1-binding site, as Runx1 targets. We then compared the predicted Runx1 targets from distinct scATAC-seq imputed matrices with differentially expressed genes after Runx1 over-expression (true labels). All imputation methods obtained higher AUPR values than the use of a raw matrix, while scOpen obtained the highest AUPR (Fig. 4h; Supplementary Fig. 12c). Among others, scOpen predicted *Tgfbr1* and *Twist2* as prominent Runx1 target genes (Fig. 4i; Supplementary Fig. 12d). We observed several peaks with high peak-to-gene correlation, increasing accessibility upon myofibroblast differentiation and presence of Runx1-binding sites. The positive impact of imputation was clear when observing scatter plots contrasting gene activity and peak accessibility of these peak-to-gene links (Fig. 4j; Supplementary Fig. 12e–i).

Another interesting question is the association of the predicted link with distinct regulatory features. While we observed no clear association of the correlation of predicted links with the size of the link (Supplementary Fig. 13a), our analysis suggested that links associated with active kidney enhancers have a higher correlation than other active regulatory regions. This further supports the functional relevance of predicted links. These results suggest that Runx1 is an important regulator of myofibroblast differentiation by regulating the EMT-related TF Twist2 and by amplifying TGFB signaling by increasing the expression of a TGFB receptor 1 and affecting the expression of extracellular matrix genes. Altogether, these results uncover a complex cascade of regulatory events across cells during the progression of fibrosis and reveal a yet unknown function of Runx1 in myofibroblast differentiation in kidney fibrosis.

## Discussion

In ATAC-seq, Tn5 generates a maximum of 2 fragments per cell in a small (~200 bp) OC region. Subsequent steps of the ATAC-seq protocol cause loss of a large proportion of these fragments. For example, only DNA fragments with the two distinct Tn5 adapters, which are only present in 50% of the fragments, are amplified in the PCR step[48]. Further DNA material losses occur during single-cell isolation, liquid handling, sequencing, or by simple financial restrictions of sequencing depth. Assuming that 25% of accessible DNA can be successfully sequenced, we expect that 56% of accessible chromatin sites will not have a single digestion event causing the so-called dropout events, assuming that digestion events follow a binomial distribution. Despite this major signal loss, imputation and denoising have been widely ignored in the scATAC-seq literature[5,6,8,9,12,13] and common scATAC-seq pipelines (e.g., Signac[49] and ArchR[34]).

We demonstrated here that scOpen estimated matrices have a higher recovery of dropout events and also improved distance and clustering results when compared to imputation methods for scRNA-seq[14,17–19,25] and the few available imputations methods tailored for scATAC-seq (cisTopic-impute[10], SCALE[20]). scOpen also presented very good scalability with the lowest memory requirements and tractable computational time on large data sets. From a methodological perspective, scOpen is the only method

performing regularization of estimated models to prevent overfitting. This is in line with a previous study, which indicated overfitting as one of the largest issues on scRNA-seq imputation[50]. Moreover, it is also possible to use the scOpen factorized matrix as a dimension reduction. We have shown that both dimensions reduced and imputed matrices from scOpen displayed the best performance on distance representation and clustering when compared to diverse state-of-the-art scATAC-seq dimension reduction/clustering pipelines (cisTopic, SnapATAC, and Cusanovich et al. 2018). Of note, the ArchR pipeline is equivalent to Cusanovich et al. 2018 and based on the same dimension reduction method (LSI). It is worth noting that LIGER[51] is another method of employing NMF for single-cell ATAC-seq analysis. It uses integrative NMF to extract shared factors with the objective of multi-modal data integration of scATAC-seq and scRNA-seq. Moreover, denoising in bulk ATAC-seq has also been approached with the use of deep learning methods[52]. These closely related approaches, however, have distinct applications than scOpen and are therefore not evaluated here.

Finally, we have demonstrated that the use of scOpen-corrected matrices improves the accuracy of existing state-of-art scATAC-seq methods (cisTopic[10], chromVAR[12], Cicero[13]). Particularly positive results were obtained in the prediction of chromatin conformation with Cicero, where all methods perform better than raw matrices. Cicero works by measuring the correlation between pairs of proximal links. Due to the fact that dropout events are independent for two regions, it is not surprising that imputation has strong benefits. This is equivalent to observations from van Dijk et al. 2018[14] in the context of scRNA-seq, where the prediction of gene–gene interactions after MAGIC imputation was significantly improved. Altogether, these results support the importance of dropout event correction with scOpen in any computational analysis of scATAC-seq. Of note, a sparsity similar to scATAC-seq is also expected in single-cell protocols based on DNA enrichment such as scChIP-seq[53,54], scCUT&Tag[55], or scBisulfite-seq[56]. Denoising and imputation of count matrices from these protocols represent a future challenge.

Moreover, we used scOpen to characterize complex cascades of regulatory changes associated with kidney injury and fibrosis. Our analyses demonstrated that a major expanding population of cells, i.e. injured PTs, myofibroblasts, and immune cells, share regulatory programs, which are associated with cell de-/differentiation and proliferation. Of all methods evaluated, scOpen obtained the best clustering results in the kidney cell repertoire using a scRNA-seq on the same kidney injury model as a reference. Trajectory analysis identified Runx1 as the major TF driving myofibroblast differentiation, which was validated by Runx1 staining in the mouse model and by retroviral over-expression studies in human PDGFRb+ kidney cells. Computational prediction with peak-to-gene links combined with footprint-supported Runx1-binding sites indicated the role of Runx1 in the regulation of *Tgfbr1* and *Twist2*. These were validated on over-expression experiments in human fibroblasts. Altogether, results suggest that Runx1 makes fibroblasts more sensitive to TGFB signaling via increasing expression of the TGFB receptors.

Runx1 has recently been reported as a potential inducer of EMT in PT cells[57]. Furthermore, in vitro data of mesenchymal stem cells (MSCs) isolated from bone marrow or prostate gland points towards a potential myofibroblast differentiation role of Runx1[58]. In vivo evidence for a functional role of Runx1 in regulating fibrogenesis has been demonstrated in zebrafish[59]. Single-cell RNA-seq data from zebrafish heart after cryo-injury suggests that endocardial cells and thrombocytes up-regulate Runx1 while Runx1 mutant zebrafish demonstrated enhanced cardiac regeneration after cryoinjury with an ameliorated fibrotic response. Here we show for the first time in vivo and in vitro

evidence that Runx1 in myofibroblasts regulates scar formation following a fibrogenic kidney injury in mice. Runx1 deficiency caused reduced myofibroblast formation and enhanced recovery. To this end, inhibiting Runx1 could lead to reduced myofibroblast differentiation and increased endogenous repair after fibrogenic organ injuries in the kidney and heart. Our results shed light on mechanisms of myofibroblasts differentiation driving kidney fibrosis and chronic kidney disease (CKD). Altogether, this demonstrates how scOpen can be used to dissect complex regulatory processes by footprinting analysis combined with peak-to-gene link predictions.

## Methods

**scOpen.** scOpen aims to simultaneously impute and reduce the dimension of a scATAC-seq matrix. Let $\mathbf{X} \in \mathbb{R}^{m \times n}$ be the scATAC-seq matrix, where $x_{ij}$ is the number of cutting sites in peak $i$ and cell $j$; $m$ is the total number of peaks and $n$ is the number of cells. We first define a binary open/closed chromatin matrix $\hat{\mathbf{X}} \in \{0,1\}^{m \times n}$, i.e.

$$x'_{ij} = \begin{cases} 1 & x_{ij} > 0 \\ 0 & x_{ij} = 0 \end{cases} \quad (1)$$

where 1 indicates the peak $i$ is open and 0 indicates closed in cell $j$. Next, we calculate a score for peak $i$ and cell $j$ by applying term frequency-inverse document frequency (TF-IDF) transformation:

$$x''_{ij} = \frac{x'_{ij}}{\sum_{p=1}^{m} x'_{pj}} \cdot \log\left(\frac{n}{\sum_{q=1}^{n} x'_{iq}}\right) \quad (2)$$

This score represents how important the peak $i$ is for cell $j$. Next, we normalize the TF-IDF matrix as

$$v_{ij} = \frac{x''_{ij}}{\sqrt{\sum_{k=1}^{m} x''^{2}_{kj}}} \quad (3)$$

We next impute the matrix by minimization of the following optimization problem:

$$\hat{\mathbf{M}} = \operatorname{argmin} \sum_{i,j} (m_{ij} - v_{ij})^2 + \lambda ||\mathbf{M}||_*, \quad \text{s.t.} \quad m_{ij} \geq 0 \quad (4)$$

where $||\mathbf{M}||_* = \sum_{i}^{k} \sigma_i(\mathbf{M})$ is the nuclear norm of matrix $\mathbf{M}$, and $\sigma_i$ denotes the $i$th largest singular value of $\mathbf{M}$. The first item is the estimator of square loss for each element in $\mathbf{M}$ and $\lambda$ is the regularization parameter, which aims to prevent the model from over-fitting and set to 1 as default value. To solve this problem, we assume that $\mathbf{M}$ is a low-rank matrix with rank $k$ and it can be written as

$$\min_{\mathbf{W},\mathbf{H}} f(\mathbf{W},\mathbf{H}) = \sum_{ij} ((\mathbf{WH})_{ij} - v_{ij})^2 + \frac{\lambda}{2} ||\mathbf{W}||^2 + \frac{\lambda}{2} ||\mathbf{H}||^2, \quad \text{s.t.} \quad (\mathbf{WH})_{ij} \geq 0 \quad (5)$$

where $\mathbf{W} \in \mathbb{R}^{m \times k}$, $\mathbf{H} \in \mathbb{R}^{k \times n}$. This constrained optimization problem is solved by using CCD methods[60]. This method iteratively updates the variable $w_{it}$ in $\mathbf{W}$ to $z$ by solving the following one-variable sub-problem:

$$\min_{z} f(z) = \sum_{j=1}^{n} \left( \left( \sum_{t' \in k} w_{it'} h_{t'j} - w_{it} h_{tj} \right) + z h_{tj} - v_{ij} \right)^2 + \frac{\lambda}{2} z^2, \quad \text{s.t.} \quad z \geq 0 \quad (6)$$

Likewise, the elements in $\mathbf{H}$ can be updated with a similar update rule. The above iteration is carried out until a termination criterion is met, e.g. number of iteration performed. Afterward, we calculate $\mathbf{M}$ as the product of $\mathbf{W}$ and $\mathbf{H}$ to obtain the scOpen imputed matrix or consider $\mathbf{H}$ as scOpen reduced matrix. This algorithm has a theoretical time complexity of $O((m+n)k)$ for a single iteration and thus is scalable for large datasets.

**Selection of hyper-parameters in scOpen.** There are two hyper-parameters in scOpen, i.e., the rank of the matrix $k$ and regularization parameter $\lambda$. Rank $k$ determines the intrinsic dimensions of a matrix and thus is highly dataset-specific. To select an appropriate value of $k$, we first input a number of ranks and generated a residual sum of squares (RSS) curve in a pre-defined interval (2–30 as default). Next, we use a knee point detection method[23], which finds a $k$ with the best trade-off between fit error and model complexity. We make use of a simulated scATAC-seq dataset as described below to evaluate the impact of $\lambda$ and $k$ in either the imputation and clustering performance (Supplementary Fig. 1). We use optimal settings ($\lambda = 1$) and knee detection from an interval of 2–30 in further results.

**scATAC-seq simulation dataset.** To generate a simulation scATAC-seq dataset, we downloaded bulk ATAC-seq data of 13 FACS-sorted human primary blood cell types from gene expression omnibus (GEO) with accession number GSE74912[61]. For each cell type, we processed the data similarly as in ref. [12]. First, the downloaded files were converted to FastQ using the SRA toolkit (http://ncbi.github.io/ sra-tools/). Next, adapter sequences and low-quality ends were trimmed from FastQ files using Trim Galore[62]. Reads were mapped to the genome hg19 using Bowtie2[63] with the following parameters ($-X\ 2000\ --very-sensitive\ --no\ -discordant$), allowing paired-end reads of up to 2 kb to align. Then, reads mapped to chrY, mitochondria and unassembled "random" contigs were removed. Duplicates were also removed with Picard and reads were further filtered for alignment quality of $> Q30$ and required to be properly paired using SAMtools[64]. Peaks were called using MACS2[65] with the following parameters ($--keep-dup\ auto\ --call-summits$). We next merged the peaks from all cell types to create a unique peaks list. We then created a peak cell-type matrix by offsetting +4 bp for forward strand and −5 bp for the reverse strand to represent the cleavage event center[1,4] and counting the number of reads start sites per cell type in each peak. This provides a cell type vs. peak matrix $\mathbf{A}$, where $a_{ij}$ indicates the number of reads for cell $j$ in peak $i$.

We next used this bulk ATAC-seq counts matrix $\mathbf{A}$ to simulate a scATAC-seq counts matrix $\mathbf{X}$ by improving the simulation strategy proposed in ref. [21]. Specifically, given $m$ peaks and $T$ cell types, we define the accessibility $x_{ij} \in \{0,1\}$ of a single cell $j$ from the cell type $t$ in peak $i$ as

$$
\begin{aligned}
x_{ij} &\sim \text{Bernoulli}\ (p_i^t) \\
p_i^t &= r_i^t \cdot n_j \cdot (1-q) + \left(\frac{1}{m}\right) \cdot n_j \cdot q \\
r_i^t &= \frac{a_{it}}{\sum_{k=1}^{m} a_{kt}} \\
n_j &= N_j \cdot f \\
N_j &\sim \text{NB}(r,p)
\end{aligned}
\quad (7)
$$

where $p_i^t$ denotes the probability of cell type $t$ being accessible in peak $i$, $q$ is a noise parameter, $n_j$ denotes the number of reads in peaks for single-cell $j$, $f$ denotes the fraction of reads in peaks (FRiP) and $N_j$ denotes the total number of reads for cell $j$. $N_j$ is sampled from a negative binomial distribution, whose parameters were estimated from a real scATAC-seq dataset. We simulated 200 cells per cell type using this process and used noise $q = 0.6$ and FRiP $f = 0.3$. Our approach differs from ref. [21] by sampling the number of reads per cell from a negative binomial distribution rather than using a fixed number and the introduction of the FRiP parameters.

**scATAC-seq benchmarking datasets.** The cell line dataset was obtained by combining single-cell ATAC-seq data of BJ, H1-ESC, K562, GM12878, TF1, and HL-60 from ref. [5], which was downloaded from GEO with accession number GSE65360. The hematopoiesis dataset includes scATAC-seq experiments of sorted progenitor cells populations: hematopoietic stem cells (HSC), multipotent progenitors (MPP), lymphoid-primed multipotent progenitors (LMPP), common myeloid progenitors (CMP), common lymphoid progenitors (CLP), granulocyte–macrophage progenitors (GMP), megakaryocyte–erythroid progenitors (MEP), and plasmacytoid dendritic cells (pDC)[6]. Sequencing libraries were obtained from GEO with accession number GSE96769. In both datasets, the original cell types were used as true labels for clustering as in the previous work[9,10]. The T cell dataset is based on human Jurkat T cells, memory T cells, naive T cells, and Th17 T cells obtained from GSE107816[8]. Labels of memory, naive, and Th17 T cells were provided in Satpathy et al. [8] by comparing scATAC-seq profiles with bulk ATAC-seq of corresponding T cell sub-populations.

For each of these three datasets, we only kept cells with at least 500 unique fragments. We then created a pseudo-bulk ATAC-seq library by merging the obtained scATAC-seq profiles and called peaks using MACS2[65]. The peaks were extended ± 250 bp from the summits as in ref. [1] and peaks overlapping with ENCODE blacklists (http://mitra.stanford.edu/kundaje/akundaje/release/blacklists/ hg19-human/) were removed. we next constructed a peak by cell counts matrix. To the test scalability of imputation methods, we also included a multiome PBMC dataset with 10,000 cells (https://support.10xgenomics.com/single-cell-multiome-atac-gex/datasets). This dataset was generated using the Chromium Single Cell Multiome ATAC + Gene Expression assay. We use here the cell types as annotated by the 10X Genomics R&D team using only the scRNA-seq data. See Supplementary Table 1 for complete statistics associated with these data sets.

**Comparison between scOpen and competing imputation methods.** We compared the performance of scOpen with 8 competing imputation approaches, i.e., MAGIC[14], SAVER[19], scImpute[17], DCA[18], cisTopic[10], scBFA[25], SCALE[20], and imputePCA. We performed imputation with these algorithms (see details below) on the benchmarking datasets.

We first tested if the imputed matrix can recover the true signal for every single cell. To this end, we used cell labels from each dataset to aggregate the ATAC-seq profiles and performed peak calling to find cell-specific OC regions. OC regions present in a particular cell type were considered as trues and OC regions not present in that cell type as negatives. For a particular cell, we can obtain true positives and true negatives by comparing the labels of the corresponding cell type with the presence of reads (or openness score) in that OC region and single cell. We use these statistics to measure the area under the precision-recall curve (AUPR)[66] for each cell.

Next, we evaluated the imputed matrix using the mean silhouette score of cells[67]. For a given cell $x$:

$$\text{silhouette}(x) = \frac{b(x) - a(x)}{\max(a(x), b(x))} \tag{8}$$

where $a(s)$ is the average distance between $x$ and the other cells of the same class, and $b(x)$ is the average distance between $x$ and cells in the closest different class. The distance was calculated as a $1 - $Pearson correlation. A higher silhouette score indicates a higher similarity of a cell to cells of the same cell type than cells from other cell types.

We next tested if the imputed matrix improves cell clustering. We applied PCA (50 PCs) for each of the imputed matrices and clustered cells using $k$-medoids and hierarchical clustering methods with $1 - $Pearson correlation as distance. Besides PCA, we also used t-SNE[68] embedding as input and euclidean as distance, as this approach is explored by cisTopic[10]. We also tested the different numbers of clusters, e.g. $k$ and $k + 1$, where $k$ is the true number of clusters. We used the ARI to evaluate the clustering results[27] with labels from benchmarking data sets (see Supplementary Fig. 1 for experimental design). The ARI measures similarity between two data clustering results by correcting the chance of grouping elements. Specifically, given two partitions of a dataset $D$ with $n$ cells, $U = \{U_1, U_2, \cdots U_r\}$ and $V = \{V_1, V_2, \cdots, V_s\}$, the number of common cells for each cluster $i$ and $j$ can be written as

$$c_{ij} = |U_i \cap V_j| \tag{9}$$

where $i \in \{1, 2, \cdots, r\}$ and $j \in \{1, 2, \cdots, s\}$. The ARI can be calculated as follows:

$$ARI = \frac{\sum_{ij} \binom{c_{ij}}{2} - \left[ \sum_i \binom{a_i}{2} \sum_j \binom{b_j}{2} \right] / \binom{n}{2}}{\frac{1}{2} \left[ \sum_i \binom{a_i}{2} + \sum_j \binom{b_j}{2} \right] - \left[ \sum_i \binom{a_i}{2} \sum_j \binom{b_j}{2} \right] / \binom{n}{2}} \tag{10}$$

where $a_i = \sum_{j=1}^{s} c_{ij}$ and $b_j = \sum_i^r c_{ij}$, respectively. The ARI has a maximum value 1 and an expected value of 0, with 1 indicating that the data clustering are exactly same and 0 indicating that the two data clustering agree randomly.

We describe below details of running all competing methods.

MAGIC: MAGIC is an algorithm for alleviating sparsity and noise of single-cell data using diffusion geometry[14]. We downloaded MAGIC from https://github.com/KrishnaswamyLab/MAGIC and applied it to the count matrix with the default setting. Prior to MAGIC, the input was normalized by library size and root squared, as suggested by the authors[14].

SAVER: SAVER is a method that recovers the true expression level of each gene in each cell by borrowing information across genes and cells[19]. We obtained SAVER from https://github.com/mohuangx/SAVER and ran it on the normalized tag count matrix with the default parameters.

scImpute: scImpute is a statistical method to accurately and robustly impute the dropouts in scRNA-seq data[17]. We downloaded scImpute from https://github.com/Vivianstats/scImpute and executed it using the default setting except for the number of cell clusters which is used to determine the candidate neighbors of each cell by scImpute. We defined this as the true cluster number for each benchmarking dataset.

DCA: DCA is a deep auto-encoder network for denoising scRNA-seq data by taking the count structure, over-dispersed nature, and sparsity of the data into account[18]. We obtained DCA from https://github.com/theislab/dca and ran it with the default setting.

cisTopic-impute: cisTopic is a probabilistic model to simultaneously identify cell states (topic-cell distribution) and cis-regulatory topics (region-topic distribution) from single-cell epigenomics data[10]. We downloaded it from https://github.com/aertslab/cisTopic and ran it with different numbers of topics (from 5 to 50). The optimal number of topics was selected based on the highest log-likelihood as suggested in ref. [10]. We then multiplied the topic-cell and the region-topic distributions to obtain the predictive distribution[10], which describes the probability of each region in each cell and is used as the imputed matrix for clustering and visualization. We termed this method cisTopic-impute.

scBFA: scBFA is a detection-based model to remove technical variation for both scRNA-seq and scATAC-seq by analyzing feature detection patterns alone and ignoring feature quantification measurements[25]. We obtained scBFA from https://github.com/quon-titative-biology/scBFA and ran it on the raw count matrix using default parameters.

SCALE: SCALE combines the variational auto-encoder (VAE) and the Gaussian mixture model (GMM) to model the distribution of high-dimensional sparse scATAC-seq data[20]. We downloaded it from https://github.com/jsxlei/SCALE and ran it with the default setting. We used option impute to get the imputed data.

imputePCA: We also included the principal component-based imputation method (termed here as imputePCA) on incomplete data sets as a control for comparison. This method is based on an interactive and regularized PCA algorithm to predict missing entries, which are considered latent variables[26]. We installed R package missMDA and performed imputation with function imputePCA with default settings. All zero entries were considered as missing data.

**Evaluation of time and memory requirement of imputation methods**. To compare the memory and running time requirement of each imputation method, we ran all of them on a dedicated HPC node with the same computation resources quota, i.e., 180 GB memory, 120 h time, and 4 CPUs. For DCA and SCALE, two deep learning-based methods, we used GPU with 16 GB memory. We measured the max memory usage during the running of a method and observed that all methods but imputePCA, SAVER, and DCA can successfully generate the imputed matrix for all datasets (Fig. 1b). For the multi-omics PBMC dataset, DCA failed due to a GPU memory issue and we could not obtain results from imputePCA and SAVER after 120 h of running.

**Comparison between scOpen and competing dimension reduction methods**. We next compared the performance of scOpen with cisTopic[10], SnapATAC[11], and latent semantic indexing (LSI) (termed here as Cusanovich2018)[69] for dimension reduction of scATAC-seq data. We applied these methods to obtain a low-dimension matrix from each dataset (detailed below) and measured the mean silhouette score[67] (see Fig. 1e).

cisTopic: We downloaded it from https://github.com/aertslab/cisTopic and ran it with different numbers of topics (from 5 to 50). The optimal number of topics was selected based on the highest log-likelihood as suggested in ref. [10]. The topic-cell distribution is used as a dimension-reduced matrix.

SnapATAC: SnapATAC is a software package for analyzing scATAC-seq datasets[11]. Instead of using peak annotation as features, it resolves cellular heterogeneity by directly comparing the similarity in genome-wide accessibility profiles between cells. Furthermore, SnapATAC uses the Nyström method to generate a low-rank embedding for a large-scale dataset which enables the analysis of scATAC-seq up to a million cells. We installed SnapATAC from https://github.com/r3fang/SnapATAC and followed the tutorial from https://github.com/r3fang/SnapATAC/blob/master/examples/10X_brain_5k/README.md to perform dimension reduction for benchmarking datasets.

Cusanovich2018: Cusanovich2018 first segments the genome into 5kb windows and then scored each cell for any insertions in these windows, generating a large, sparse, binary matrix of 5 kb windows by cells. Based on this matrix, the top 20,000 most commonly used sites were retained. Then, the matrix was normalized and re-scaled using the term frequency-inverse document frequency (TF-IDF) transformation. Next, singular value decomposition (SVD) was performed to generate a PCs-by-cells low dimension matrix.

**Benchmarking of scATAC-seq downstream analysis methods**. Next, we compared the performance of state-of-art scATAC-seq methods (scABC, chromVAR, and Cicero) when presented with either scOpen imputed or raw scATAC-seq matrix. The rationale is if we improve the count matrix by imputation, we should be able to improve downstream analysis. Note that scABC is the only method providing a clustering solution. chromVAR and Cicero transform the scATAC-seq matrices into TF and gene space. We here again evaluated the results based on clustering accuracy with methods used as the standard by these pipelines, i.e., hierarchical clustering with complete agglomeration method for chromVAR and $k$-medoids for Cicero. Moreover, we evaluated the co-accessible links predicted by Cicero between using scOpen imputed or raw counts matrix.

scABC: scABC is an unsupervised clustering algorithm for single-cell epigenomic data[9]. We downloaded it from https://github.com/SUwonglab/scABC and executed it according to the tutorial https://github.com/SUwonglab/scABC/blob/master/vignettes/ClusteringWithCountsMatrix.html.

chromVAR: chromVAR is an R package for analyzing sparse chromatin-accessibility data by measuring the gain or loss of chromatin accessibility within sets of genomic features, as regions with sequence predicted TF-binding sites[12]. We obtained chromVAR from https://github.com/GreenleafLab/chromVAR and executed to find gain/loss of chromatin accessibility in regions with binding sites of 571 TF motifs obtained in JASPAR version 2018[70].

Cicero: Cicero is a method that predicts co-accessible pairs of DNA elements using single-cell chromatin accessibility data[13]. Moreover, Cicero provides a gene activity score for each cell and gene by assessing the overall accessibility of a promoter and its associated distal sites. This matrix was used for clustering and visualization of scATAC-seq. We obtained Cicero from https://github.com/cole-trapnell-lab/cicero-release and executed it according to the document provided by https://cole-trapnell-lab.github.io/cicero-release/docs/.

**Chromosomal conformation experiments with Cicero**. We used conformation data as true labels to evaluate co-accessible pairs of cis-regulatory DNA as detected by Cicero on GM12878 cells. We obtained a scATAC-seq matrix of GM12878 cells from GEO (GSM2970932). For evaluation, we downloaded promoter-capture (PC) Hi-C data of GM12878 from GEO (GSE81503), which uses the CHiCAGO[71] score as a physical proximity indicator. We also downloaded ChIA-PET data of GM12878 from GEO (GSM1872887), which used the frequency of each interaction PET cluster to represent how strong the interaction is. We considered all obtained links, as provided by these data sets, as true interactions as in ref. [13]. To investigate the performance of each method against the number of cells, we also randomly down-sampled the data to 50% and 25%. Next, we replicated the evaluation analysis performed in Fig. 4 of ref. [13] and contrasted the results of Cicero with raw or

matrices obtained after scOpen imputation. Next, we use the built-in function *compare_connections* of Cicero to define the true labels for predicted co-accessibility links. Using the correlation as prediction, we finally computed the area of precision and recall curve (AUPR) with *pr.curve* function from R package PRROC[72].

**Unilateral ureter obstruction (UUO) mouse kidney animal experiments**. UUO was performed as previously described[30]. Shortly, the left ureter was tied off at the level of the lower pole with two 7.0 ties (Ethicon) after flank incision. One C57BL/6 male mouse (age 8–10 weeks) was sacrificed on day 0 (sham), day 2, and 10 after the surgery. Kidneys were snap-frozen immediately after sacrifice. Pdgfrb-BAC-eGFP reporter mice (for staining experiments, age 8–12 weeks, C57BL/6) were developed by N. Heintz (The Rockefeller University) for the GENSAT project. Genotyping of all mice was performed by PCR. Mice were housed under specific pathogen-free conditions at the University Clinic Aachen. Pdgfrb-BAC-eGFP were sacrificed on day 10 after the surgery. All animal experiment protocols were approved by the LANUV-NRW, Düsseldorf, Germany. All animal experiments were carried out in accordance with their guidelines.

**UUO scATAC-seq experiments**. Nuclei isolation was performed as recommended by 10X Genomics (demonstrated protocol CG000169). The nuclei concentration was verified using stained nuclei in a Neubauer chamber with trypan-blue targeting a concentration of 10,000 nuclei. Tn5 incubation and library prep followed the 10X scATAC protocol. After quality check using Agilent BioAnalyzer, libraries were pooled and run on a NextSeq in 2 × 75 bps paired-end run using three runs of the NextSeq 500/550 High Output Kit v2.5 Kit (Illumina). This results in more than 600 million reads.

**UUO data pre-processing**. We used Cell-Ranger ATAC (v1.1.0) pipeline to perform low-level data processing (https://support.10xgenomics.com/single-cell-atac/software/pipelines/latest/algorithms/overview). We first demultiplexed raw base call files using cellranger-atac mkfastq with its default setting to generate FASTQ files for each flowcell. Next, cellranger-atac count was applied to perform read trimming, filtering, and alignment. We then estimated the transcription start site (TSS) enrichment score using the obtained fragment files and filtered low-quality cells using a TSS score of 8 and a number of unique fragments of 1000 as thresholds. The obtained barcodes are considered valid cells for the following analysis.

**UUO data dimension reduction, data integration, and clustering**. We next performed peak calling using MACS2 for each sample and merged the peaks to generate a union peak set, which was used to create a peak by cell matrix. For comparison, we applied distinct methods, i.e., scOpen, cisTopic, SnapATAC, and LSI/Cusanovich2018, to the matrix and used the dimension reduced matrix for data integration, clustering, and visualization. Next, we used Harmony[31] to integrate the scATAC-seq profiles from different conditions (day 0, day 2, and day 10) using either LSI/Cusanovich2018, cisTopic, scOpen, or SnapATAC dimension reduced matrix as input. Specifically, we created a Seurat object for each of the low-dimension matrices and ran the Harmony algorithm with the function *RunHarmony*. We then used k-medoids to cluster the cells taking batch-corrected low-dimension matrix as input. The number of clusters was set to 17 given that the single-nucleus RNA-seq that we used as a reference for annotation identified 17 unique cell types (see below).

**UUO label transfer**. To evaluate and annotate the clusters obtained from data integration, we downloaded a publicly available snRNA-seq dataset of the same fibrosis model (GSE119531) and performed label transfer using Seurat3[33]. This dataset contains 6147 single-nucleus transcriptomes with 17 unique cell types[32]. For label transfer, we used the gene activity score matrix estimated by ArchR and transferred the cell types from the snRNA-seq dataset to the integrated scATAC-seq dataset by using the function *FindTransferAnchors* and *TransferData* in Seurat3[33]. For benchmarking purposes, the predicted labels were used as true labels to compute ARI for evaluation of the clustering results and silhouette score for evaluation distances after using different dimension reduction methods as input for data integration (Supplementary Fig. 7c–e). We also performed the same analysis for each sample separately and evaluated the results (Fig. 3a).

**UUO cluster annotation**. For the biological interpretation, we estimated doublet scores using ArchR[34] and removed cells with a doublet score higher than 2.5. Next, we named the cluster by assigning the label with the highest proportion of cells to the cluster and checking marker genes (Supplementary Fig. 9a). In total, we recovered 16 unique cell types from the 17 labels, as two clusters (2 and 17) were annotated as TAL cells. Specifically, we denoted clusters 6, 1, 3 as proximal tubule (PT) S1, S2, and S3 cells. We annotated cluster 2 as thick ascending limb (TAL), cluster 5 as distal convoluted tubule (DCT), cluster 7 as collecting duct-principal cell (CD-PC), cluster 8 as an EC, cluster 9 as connecting tubule (CNT), cluster 10 as an intercalated cell (IC), cluster 11 as fibroblast, cluster 12 as descending limb + thin ascending limb (DL and TAL), cluster 13 as macrophage (MAC), cluster 16 as

podocytes (Pod). Cluster 14 was identified as injured PT, which was not described in ref. [32], given the increased accessibility of marker *Vcam1* and *Havcr1* (Supplementary Fig. 9a). We also renamed the cells of cluster 15, which were label as Mac2 in ref. [32], as lymphoid cells given that these cells express B and T cell markers *Ltb* and *Cd1d*, but not macrophage markers *C1qa* and *C1qb*. Finally, cluster 4 was removed based on the doublet analysis.

**UUO Cell-type-specific footprinting with HINT-ATAC**. We have adapted the footprinting-based differential TF activity analysis from HINT-ATAC for scATAC-seq. In short, we created pseudo bulk atac-seq libraries by combining reads of cells for each cell type and performed footprinting with HINT-ATAC. Next, we predicted TF binding sites by motif analysis (FDR = 0.0001) inside footprint sequences using RGT (v0.12.3; https://github.com/CostaLab/reg-gen). Motifs were obtained from JASPAR Version 2020[73]. We measured the average digestion profiles around all binding sites of a given TF for each pseudo-bulk ATAC-seq library. We used then the protection score[4], which measures the cell-specific activity of a factor by considering the number of digestion events around the binding sites and depth of the footprint. Higher protection scores indicate higher activity (binding) of that factor. Finally, we only considered TFs with more than 1000 binding sites and variance in activity score higher than 0.3. We also performed smoothing for visualization of average footprint profiles. In short, we performed a trimmed mean smoothing (5 bps window) and ignored cleavage values in the top 97.5% quantile for each average profile.

**Identifying trajectory from fibroblast to myofibroblast**. We performed further sub-clustering of fibroblast cells on batch-corrected low-dimension scOpen matrix. In total, 3 clusters were obtained and annotated as pericyte (cluster 1), myofibroblast (cluster 2), and Scara5+ fibroblast (cluster 3) using known marker genes (Supplementary Fig. 10a), respectively. For visualization, a diffusion map 2D embedding was generated using R package density[74]. Next, a trajectory from Scara5+ fibroblast to myofibroblast was created using function *addTrajectory* and visualized using function *plotTrajectory* from ArchR (Supplementary Fig. 10c).

**Identifying key TF drivers of myofibroblast differentiation**. To identify TFs that drive this process, we first performed peak calling based on all fibroblasts using MACS2 to obtain specific peaks and then estimated motif deviation per cell using chromVAR. The deviation scores were normalized to allow for comparison between TFs. Next, we selected the TFs with high variance of deviation and gene activity score along the trajectory and calculated the correlation of TF activity and gene activity. This was done by the function *correlateTrajectories* from ArchR. We only consider the 31 TFs with significant correlation (FDR < 0.1) (Fig. 4c). We then sorted the TFs by correlation, which identifies Runx1 as the most relevant TF for the differentiation (Supplementary Fig. 10d).

**Prediction of peak-to-gene links**. We obtained TSS from annotation BSgenome.Mmusculus.UCSC.mm10 for each gene and extended it by 250 kbps for both directions. Then, we overlapped the peaks from fibroblasts and the TSS regions using function *findOverlaps* to identify putative peak-to-gene links. We next created 100 pseudo-bulk ATAC-seq profiles by assigning each cell to an interval along the trajectory of myofibroblast differentiation. The gene score matrix and peak matrix were aggregated according to the assignment to generate two pseudo-bulk data matrices. For each putative peak-to-gene link, we calculated the correlation between peak accessibility and gene activity. The p-values are computed using t distribution and corrected by Benjamini–Hochberg method. For comparison, we also performed matrix imputation using the four top methods, i.e., scOpen, SCALE, MAGIC, and cisTopic, as evaluated by peaks recovering (Supplementary Fig. 2b) and computed the correlation based on the imputed matrix.

To compare the scOpen predicted peak-to-gene correlation from different types of peaks, we used the annotation generated by R package ArchR[34] and classified the peaks as distal, exonic, intronic, and promoter. We also tested if the correlation is different between activate enhancers and nonenhancers. For this, We obtained H3K27ac (ENCSR000CDG) and H3K4me1 (ENCSR436FYE) ChIP-seq peaks of mouse kidneys from ENCODE. The peaks were classified as active enhancers if they are overlapping with H3K27Ac and H3K4me1, and other active regions if they are only overlapping with H3K27Ac.

**Prediction and evaluation of Runx1 target genes**. With each peak being associated with genes, we next sought to link Runx1 to its target genes. For this, we first performed a footprinting pseudo-bulk ATAC-seq profile to identify TF footprints inside peaks linked to genes in the previous peak-to-gene analysis. Next, we identified Runx1-binding sites using a motif-matching approach. We defined the genes that have at least one footprint-support binding site of Runx1 in their associated peaks as Runx1 target genes. We then used the peak-to-gene correlation as a prediction between Runx1 and the target genes. This procedure was for the peak to gene links predicted by distinct imputation approaches, thus generating various predictions. To evaluate the results, we used the DE genes obtained from RNA-seq of Runx1 overexpression as true labels (see below), and computed the AUPR (Fig. 4h).

**Immunofluorescence staining of Runx1**. Mouse kidney tissues were fixed in 4% formalin for 2 h at RT and frozen in OCT after dehydration in 304%4 sucrose overnight. Using 5–10 μm cryosections, slides were blocked in 5% donkey serum followed by 1-h incubation of the primary antibody, washing three times for 5 min in PBS, and subsequent incubation of the secondary antibodies for 45 min. Following DAPI (4′,6-diamidino-2-phenylindole) staining (Roche, 1:10,000) the slides were mounted with ProLong Gold (Invitrogen, #P10144). Cells were fixed with 3% paraformaldehyde followed by permeabilization with 0.3% TritonX. Cells were incubated with primary antibodies and secondary antibodies diluted in 2% bovine serum albumin in PBS for 60 or 30 min, respectively. The following antibodies were used: anti-Runx1 (HPA004176, 1:100, Sigma-Aldrich), AF647 donkey anti-rabbit (1:200, Jackson Immuno Research).

**Confocal imaging and quantification**. Images were acquired using a Nikon A1R confocal microscope using ×40 and ×60 objectives (Nikon). Raw imaging data were processed using Nikon Software or ImageJ. Systematic random sampling was applied to the sub-sample of at least three representative areas per image of PDGFRbeGFP mice (n = 3 mice per condition). Using QuPath nuclei were segmented and fluorescent intensity per nuclear size was measured of PDGFRbeGFP positive nuclei.

**Ethics**. The ethics committee of the University Hospital RWTH Aachen approved the human tissue protocol for cell isolation (EK-016/17). Kidney tissues were collected from the Urology Department of the University Hospital Eschweiler from patients undergoing nephrectomy due to renal cell carcinoma.

**Generation of a human PDGFRb+ cell line**. The cell line was generated using MACS separation (Miltenyi biotec, autoMACS Pro Separator, #130-092-545, autoMACS Columns #130-021-101) of PDGFRb+ cells that were isolated from the healthy part of the kidney cortex after nephrectomy as previously described in ref. [43]. The following antibodies were used for staining the cells and MACS procedure: PDGFRb (R&D #MAB1263 antibody, dilution 1:100) and anti-mouse IgG1-MicroBeads solution (Miltenyi, #130-047-102). The cells were cultured in DMEM media (Thermo Fisher #31885) added 10% FCS and 1% penicillin/streptomycin for 14 days. For immortalization (SV40-LT and HTERT) the retroviral particles were produced by transient transfection of HEK293T cells using TransIT-LT (Mirus). Amphotropic particles were generated by co-transfection of plasmids pBABE-puro-SV40-LT (Addgene #13970) or xlox-dNGFR-TERT (Addgene #69805) in combination with a packaging plasmid pUMVC (Addgene #8449) and a pseudotyping plasmid pMD2.G (Addgene #12259) respectively. Using Retro-X concentrator (Clontech) 48 h post-transfection the particles were concentrated. For transduction, the target cells were incubated with serial dilutions of the retroviral supernatant (1:1 mix of concentrated particles containing SV40-LT or rather hTERT) for 48 h. At 72 h after transfection, the infected PDGFRb+ cells were selected with 2 μg/ml puromycin for 7 days.

**Retroviral overexpression of Runx1**. Runx1 vector construction and generation of stable Runx1-overexpressing cell lines. The human cDNA of Runx1 was PCR amplified from 293T cells (ATCC, CRL-3216) using the primer sequences 5′-atgcgtatccccgtagatgcc-3′ and 5′-tcagtagggcctccacacgg-3′. Restriction sites and N-terminal 1xHA-Tag have been introduced into the PCR product using the primer 5′-cactcgaggccaccatgtacccatacgatgttccagattacgctcgtatccccgtagatgcc-3′ and 5′-acggaattctcagtagggcctccacac-3′. Subsequently, the PCR product was digested with XhoI and EcoRI and cloned into pMIG (pMIG was a gift from William Hahn (Addgene plasmid #9044; http://www.addgene.org/9044/; RRID:Addgene_9044). Retroviral particles were produced by transient transfection in combination with packaging plasmid pUMVC (pUMVC was a gift from Bob Weinberg (Addgene plasmid #8449)) and pseudotyping plasmid pMD2.G (pMD2.G was a gift from Didier Trono (Addgene plasmid #12259; http://www.addgene.org/12259/; RRID:Addgene_12259)) using TransIT-LT (Mirus). Viral supernatants were collected 48–72 h after transfection, clarified by centrifugation, supplemented with 10% FCS and Polybrene (Sigma-Aldrich, final concentration of 8 μg/mL) and 0.45 μm filtered (Millipore; SLHP033RS). Cell transduction was performed by incubating the PDGFβ cells with viral supernatants for 48 h. eGFP-expressing cells were single-cell sorted. A complete list of primers used in this study is provided in Supplementary Table 2.

**RNA isolation, RNA-Seq library preparation, and sequencing**. RNA was extracted according to the manufacturer's instructions using the RNeasy Kit (QIAGEN). For RNA-seq Illumina TruSeq Stranded Total RNA Library Preparation Kit was used using 1000 ng RNA as input. Sequencing libraries were quantified using Tapestation (Agilent) and Quantus (Promega). Equimolar pooling of the libraries was normalized to 1.8 pM, denatured using 0.2 N NaOH, and neutralized with 200 nM Tris pH 7.0 prior to sequencing. Final sequencing was performed on a NextSeq500/550 platform (Illumina) according to the manufacturer's protocols (Illumina, CA, USA).

**Analysis of RNA-seq data**. Pipeline nf-core/rnaseq[75] was used to analyze RNA-seq data. Briefly, reads were aligned to the hg38 reference genome using STAR[76] and gene expression was quantified with Salmon[77]. Deferentially expressed genes were identified using DESeq2[78]. We used an adjusted p-value of 1e−05 and log2 fold change of 1 as thresholds to select the significant DE genes, which were used as true labels to evaluate the Runx1 target gene prediction. GO-enrichment analysis was performed R package gprofiler2 and we showed results for biological process and pathways from Human Phenotype Ontology (Supplementary Fig. 11e). The volcano plot was generated by using the R package EnhancedVolcano.

**Calculation of population-doubling level (PDL)**. For determining PDL, PDGFRb cells overexpressing Runx1 (or as control having genomically integrated the empty vector sequence) were passaged in six-well plates at a density of $1.5 \times 10(4)$ cells/well. Every 96 h (at sub-confluent state), cells were harvested and counted in a hemocytometer before being re-seeded at initial density.

**Statistical analysis and reproducibility**. All reported p-values based on multi-comparison tests were corrected using the Benjamini–Hochberg method. The number of samples for each group was chosen on the basis of the expected levels of variation and consistency. The depicted immunofluorescence micrographs are representative. All studies were performed at least three times, and all repeats were successful.

**Reporting summary**. Further information on research design is available in the Nature Research Reporting Summary linked to this article.

## Data availability

The scATAC-seq data generated from UUO mouse kidney and RNA-seq data from Runx1 over-expression of human fibroblasts in this study have been deposited in NCBI Gene Expression Omnibus and are accessible through GEO Series accession number GSE139950. Pre-processed scATAC-seq data from UUO mouse kidneys are provided in zenodo. The ATAC-seq for human primary blood cell type, cell lines, hematopoiesis, and T-cells are available in GEO under accession codes GSE74912, GSE65360, GSE96769, and GSE107816. The scATAC-seq for the PBMC dataset was downloaded from here. The scATAC-seq matrix, Hi-C, and ChIA-PET data of GM12878 are available in GEO under accession codes GSM2970932, GSE81503, and GSM1872887. The H3K27ac and H3K4me1 ChIP-seq peaks of mouse kidneys were obtained from ENCODE with experiment ENCSR000CDG and ENCSR436FYE. Source data are provided with this paper.

## Code availability

The scOpen code is available in GitHub[79] (https://github.com/CostaLab/scopen). All scripts for reproducing the analysis are available at https://github.com/CostaLab/scopen-reproducibility, as well as tables with all benchmarking results and raw count matrices from benchmarking datasets. Tutorial for the use of HINT-ATAC with the hematopoietic data set is provided in https://www.regulatory-genomics.org/hint/tutorial-differential-footprints-on-scatac-seq/.

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

## Acknowledgements

This work was funded by grants of the Interdisciplinary Center for Clinical Research (IZKF) Aachen, RWTH Aachen University Medical School, Aachen, Germany and by the Deutsche Forschungsgemeinschaft (DFG-GE 2811/3) to I.G.C. and (DFG SFB/TRR57 P30, SFB/TRR219 P5) and a Grant of the European Research Council (ERC-StG 677448) to R.K. and by the Bundesministerium für Bildung und Forschung (BMBF e:Med Consortia Fibromap) to I.G.C. and R.K. C.K. was partly funded by the clinician-scientist program of the German Society of Internal Medicine (DGIM) and a Gerok position of the DFG SFB/TRR 219, P5. Simulations were performed with computing resources granted by ITC RWTH Aachen University under projects rwth0233 and rwth0429. We thank the team of the IZKF Aachen Genomics Core facility for sequencing experiments.

## Author contributions

Z.L., I.G.C., C.K., and R.K. conceived the study; Z.L. implemented the algorithm; Z.L., M.C., and I.G.C. performed the computational analyses; C.K., S.Z., N.K., and S.M. conducted the experiments; M.Z. supported the data interpretation; Z.L., I.G.C., C.K., and R.K. wrote the paper with inputs from all other co-authors; all authors reviewed the manuscript.

## Funding

## Competing interests

The authors declare no competing interests.
