## [Peer Review File · Nature Communications]

Chromatin-accessibility estimation from single-cell ATAC data with scOpenEditorial Note: This manuscript has been previously reviewed at another journal that is not operating a transparent peer review scheme. This document only contains reviewer comments and rebuttal letters for versions considered at *Nature Communications*.

REVIEWER COMMENTS

Reviewer #1 (Remarks to the Author):

The authors have addressed all my concerns. The manuscript is now ready for publication.

Reviewer #3 (Remarks to the Author):

The authors have substantially improved their manuscript and they have shared the code to make the analysis reproducible. Overall, I believe that this study is now close to be accepted for publication. I still have a few comments which should hopefully be fast to address:

- The authors have completely changed the methodology of scOPEN. In the initial report they proposed the use of positive-unlabelled factorisation, which seemed like an interesting and novel choice for the analysis of binary scATAC-seq data. In contrast, now the authors adopt regularised NMF, which is not novel for the analysis of single-cell genomics data (see for example the LIGER software). Can the authors explain why they changed the underlying statistical model?

- Although the authors convincingly demonstrate that NMF is a good method for dimensionality reduction and imputation (and this on itself warrants publication of the study), I don't think it should be rebranded as a new method. Unless the authors have introduced key methodological innovations not present in the regularised NMF implementation, they should add the term NMF in their manuscript and/or software title to clarify the model choice.

- The fact that scOPEN is implemented in Python (but accessible from R) is an advantage, as a lot of the single-cell analysis ecosystem is transitioning to Python. However, the authors should ensure that their software displays interoperability with other popular Python-based frameworks such as scanpy. From what I can see in <https://github.com/CostaLab/scopen>. Can the authors add a notebook to explain how to operate with scanpy?

- The authors have included a comparison with PCA, but I do not understand how PCA shows worse results than using the "Raw" data. At the end of the day, SVD (Cusanovich2018) and PCA are closely related methods. Could the authors clarify?

- In the methods section, for the clustering benchmarking, the authors state "(...) We used t-SNE embedding as input and euclidean as distance." t-SNE does not preserve the global structure of the data, clustering should not be performed on t-SNE space, but rather on the (linear) PCA space. If the authors disagree, could they justify their choice?

Reviewer #4 (Remarks to the Author):

The authors have made a massive effort to greatly revise the submitted work, and I congratulate them on the vastly improved clarity and demonstrations of the utility of scOpen. In response to my original critiques, I have only a few minor comments on the manuscript/rebuttal in its current form. However, neither requires meaningful changes to the manuscript, and I would therefore suggest that the current version is suitable for publication.

1) The donor / batch effect was shown to be minimal in the analyses of these data in Buenrostro et al. 2018 (Supplemental Figure 2), so there is some concern that scOpen is finding these effects whereas the prior analytical strategy did not reveal them. However, by removing this dataset from the paper and demonstrating an analysis in the paper where scOpen can feed into batch correction approaches like harmony, the authors have adequately addressed this.

2) The authors assert that the choice of 2,000 peaks is due to its implementation in Seurat/Signac for label transfer. This is not correct. The default of Seurat utilizes 2,000 gene activity scores for CCA analysis, but for LSI, the typical threshold is "q0" or "q50", resulting in often ~100,000 peaks. However, the new analysis and benchmarking presented in Figure 1 alleviates concerns about the scalability of scOpen.

3) I appreciate the added work required to improve the online resources, which I believe will be an asset to the community.

Reviewer #5 (Expertise: investigating tissue injury responses at the single cell level):

The authors have developed and validated a new computational tool to address the dropout problem in scATAC-seq data using imputation. This tool addresses an important problem and should be useful for the field, but there are a number of issues that should be addressed to improve the paper.

1. The authors demonstrate that scOpen has increased sensitivity for detection of open chromatin regions when compared with existing tools. The comparison in supplemental fig 2 appears to group all cell types together before evaluating each method using AUPR. Was the improvement comparable across all cell types (including less common cell types)? In other words, are the benefits of imputation predominantly due to denoising/detection of peaks in well-represented cell types in the dataset?
2. The authors use annotated cell labels to evaluate the performance of scOpen relative to the other tools. There is significant interobserver variability in cell type annotation, which is a potential confounder because it assumes that cells have been properly annotated (and that two users would assign the same labels to each cell). The same problem may arise when label transfer is performed to annotate an scATAC dataset with another single cell dataset.
3. The comparison between scOpen predicted chromatin-chromatin interactions (using Cicero) and ChIA-PET and Hi-C datasets looks very encouraging. In this reviewers' experience, predicting chromatin-chromatin interactions (or gene-enhancer interactions) is a more difficult challenge than dimensional reduction or clustering, which gives comparable results across different methods. The comparison in Fig 2e,f was done with an H1-ESC cell line as opposed to a heterogeneous sample consisting of multiple cell types. Does scOpen have similar benefits in predicting chromatin-chromatin interactions for complex samples with rare cell types?
4. The authors compared scOpen to a wide variety of scATAC analysis tools, but did not include AtacWorks (Lal et al PMID: 33686069). AtacWorks is a deep learning-based tool built on the Nvidia parabricks architecture that can denoise scATAC datasets to improve peak calling (especially in rare cell types). This is a different approach that the authors should consider commenting on in their discussion.
5. NFkB signaling has been implicated in murine proximal tubule IRI and in human dedifferentiated PT (the latter based on multiomics, similar to this study). Did the authors detect any changes in NFkB TF activity in their UUO time course?
6. The authors state that they consider all genes with a TF footprint-supported Runx1 binding site as a Runx1 target. What proportion of these sites have a validated (eg. ChIP-seq or other method) Runx1 binding site in a public database like ENCODE? The correlation between peak-to-gene following scOpen imputation was modest with $R=0.4$ (Fig 4J). Did the authors observe any difference in correlation for peak-to-gene relationships depending on the location of the peak relative to the TSS. For example, did peaks located within 50kb of the TSS correlate better with respect to peak-to-gene relationships? How about distal intergenic peaks vs. proximal peaks? Or peaks that overlap with known enhancer marks (eg. H3K27Ac)?

7. The Runx1 data is presented as if it is a new finding, but this TF is already known to regulate the myofibroblast transition (PMID 25313057) including from the authors own work (BioRxiv411686). This is somewhat misleading, and this needs to be placed into context based on the prior published work.

Referee comments:

Reviewer #1 (Remark to the Author):

The authors have addressed all my concerns. The manuscript is now ready for publication.

We thank the reviewer for a very positive evaluation of our manuscript.

Reviewer #3 (Remark to the Author):

The authors have substantially improved their manuscript and they have shared the code to make the analysis reproducible. Overall, I believe that this study is now close to be accepted for publication. I still have a few comments which should hopefully be fast to address.

We appreciate the positive evaluation of our manuscript.

1) The authors have completely changed the methodology of scOPEN. In the initial report they proposed the use of positive-unlabelled factorisation, which seemed like an interesting and novel choice for the analysis of binary scATAC-seq data. In contrast, now the authors adopt regularised NMF, which is not novel for the analysis of single-cell genomics data (see for example the LIGER software). Can the authors explain why they changed the underlying statistical model?

The positive-unlabelled matrix factorization model requires a dropout rate, i.e., the fraction of non-observed elements denoted ρ . scOpen initially assumed that the dropout rate ρ is associated with the number of fragments with a maximal value of 0.5. To address a request from reviewer 2, which pointed out this assumption/parametrization seemed arbitrary, we performed parameter evaluation of the dropout and observed that this parameter impacted the results. This model simplification made scOpen equal to a regularised NMF.

We mention LIGER in the manuscript (see discussion). In short, LIGER uses integrative NMF (iNMF) to extract shared factors for multimodal data integration, while in scOpen uses regularized NMF to impute/denoise the scATAC-seq data. These methods have quite specific applications and cannot be used in each other's app.

2) The fact that scOPEN is implemented in Python (but accessible from R) is an advantage, as a lot of the single-cell analysis ecosystem is transitioning to Python. However, the authors should ensure that their software displays interoperability with other popular Python-based frameworks such as scanpy. From what I can see in <https://github.com/CostaLab/scopen>. Can the authors add a notebook to explain how to operate with scanpy?

We appreciate this suggestion. We have expanded the scOpen tutorial with a jupyter notebook (<https://github.com/CostaLab/scopen/blob/master/vignettes/epiScanpy.ipynb>), which shows how scOpen can be used with (epi)scanpy to analyze scATAC-seq data.

3) The authors have included a comparison with PCA, but I do not understand how PCA shows worse results than using the "Raw" data. At the end of the day, SVD (Cusanovich2018) and PCA are closely related methods. Could the authors clarify?

We agree that SVD and PCA are similar methods for dimension reduction. In this manuscript, SVD was used for dimension reduction within the Cusanovich 2018 scATAC-seq pipeline (results are shown in Fig. 2a-b). A PCA-based imputation method was used as a baseline following the request of review #2 in Fig. 1. The PCA imputation is based on an interactive PCA algorithm to predict missing entries, which are considered as latent variables (see Husson, 2016 doi:10.18637/jss.v070.i01). A possible reason why PCA imputation performs poorly in the high sparsity of the data. To make these points clear in the manuscript, we improved results and methods descriptions and named the method as imputePCA instead of PCA.

4) In the methods section, for the clustering benchmarking, the authors state "(...) We used t-SNE embedding as input and euclidean as distance." t-SNE does not preserve the global structure of the data, clustering should not be performed on t-SNE space, but rather on the (linear) PCA space. If the authors disagree, could they justify their choice?

Our clustering benchmarking (Fig. 1) is based on both PCA and t-SNE spaces. We have improved the method description to make this clear. We agree with the reviewer that performing clustering on t-SNE space is not an optimal choice, but t-SNE was included as it is adopted by cisTopic (see page 6 of González-Blas, et al.). We have generated a similar analysis as in Fig. 1f and Supplementary Fig. 2e by only considering benchmarking on PCA spaces (Reply Letter Fig. 1). The results are similar to those in the main manuscript: scOpen obtained the highest clustering accuracy in 3 out of 4 benchmarking datasets and is the best ranked method. This indicates that the use of t-SNE does not impact the current manuscript results. We have also improved the method description to make this point clear.

Reply Letter Fig.1. **(a)** Barplots showing the clustering accuracy for distinct imputation methods. Y-axis indicates the Adjusted Rand Index. Dots represent individual ARI values of distinct clustering methods. **(b)** Ranking of methods using ARI as the metric for benchmarking datasets. Asterisks denote statistical significance: * $p < 0.05$, *** $p < 0.001$.

Reviewer #4 (Remark to the Author):

The authors have made a massive effort to greatly revise the submitted work, and I congratulate them on the vastly improved clarity and demonstrations of the utility of scOpen. In response to my original critiques, I have only a few minor comments on the manuscript/rebuttal in its current form. However, neither requires meaningful changes to the manuscript, and I would therefore suggest that the current version is suitable for publication.

1) The donor / batch effect was shown to be minimal in the analyses of these data in Buenrostro et al. 2018 (Supplemental Figure 2), so there is some concern that scOpen is finding these effects whereas the prior analytical strategy did not reveal them. However, by removing this dataset from the paper and demonstrating an analysis in the paper where scOpen can feed into batch correction approaches like harmony, the authors have adequately addressed this.

2) The authors assert that the choice of 2,000 peaks is due to its implementation in Seurat/Signac for label transfer. This is not correct. The default of Seurat utilizes 2,000 gene activity scores for CCA analysis, but for LSI, the typical threshold is "q0" or "q50", resulting in often ~100,000 peaks. However, the new analysis and benchmarking presented in Figure 1 alleviates concerns about the scalability of scOpen.

3) I appreciate the added work required to improve the online resources, which I believe will be an asset to the community.

We appreciate the constructive and very positive comments.

Reviewer #5 (Expertise: investigating tissue injury responses at the single cell level):

The authors have developed and validated a new computational tool to address the dropout problem in scATAC-seq data using imputation. This tool addresses an important problem and should be useful for the field, but there are a number of issues that should be addressed to improve the paper.

1. The authors demonstrate that scOpen has increased sensitivity for detection of open chromatin regions when compared with existing tools. The comparison in supplemental fig 2 appears to group all cell types together before evaluating each method using AUPR. Was the improvement comparable across all cell types (including less common cell types)? In other words, are the benefits of imputation predominantly due to denoising/detection of peaks in well-represented cell types in the dataset?

This is an interesting aspect worth further investigation. We have averaged the AUPR of all cell types in all benchmarking data sets and evaluated the association of the AUPR with the number of cells for all data sets together (see Reply Letter Fig. 2). While there is an association between the (log) number of cells and mean AUPR for all methods, scOpen and MAGIC are among the methods that were less influenced by cell type size than cisTopic. We have incorporated these results in the manuscript (see Results section and **Supplementary Fig. 2c**).

Reply Letter Fig.2. Scatter plot comparing average AUPR and number of cells per cell type across all datasets for all methods. Each dot represents a cell type. The x-axis represents the number of cells for each cell type. The trend line was fitted for each method.

2. The authors use annotated cell labels to evaluate the performance of scOpen relative to the other tools. There is significant interobserver variability in cell type annotation, which is a potential confounder because it assumes that cells have been properly annotated (and that two users would assign the same labels to each cell). The same problem may arise when label transfer is performed to annotate an scATAC dataset with another single cell dataset.

We agree with the reviewer that annotating cells can be subjective and is dependent on clustering analysis. Altogether, defining labels for clustering problems is hard. To address this issue, we have carefully selected our benchmarking datasets. For the first three datasets, the cell labels were either from different scATAC-seq libraries (e.g., cell lines) or based on sorting (FACS) with cell surface markers (e.g., Hematopoiesis and T-cells). For multi-omic PBMC

data and UVO data, the cells were annotated using a distinct modality (snRNA-seq). Also, no imputation method was used in the manually annotated snRNA-seq data sets or in the label transfer process. As all imputation methods are evaluated equally under the same clustering methods, we believe no selection bias exists in our analysis.

3. The comparison between scOpen predicted chromatin-chromatin interactions (using Cicero) and ChIA-PET and Hi-C datasets looks very encouraging. In this reviewer's experience, predicting chromatin-chromatin interactions (or gene-enhancer interactions) is a more difficult challenge than dimensional reduction or clustering, which gives comparable results across different methods. The comparison in Fig 2e,f was done with an H1-ESC cell line as opposed to a heterogeneous sample consisting of multiple cell types. Does scOpen have similar benefits in predicting chromatin-chromatin interactions for complex samples with rare cell types?

We thank the reviewer for the supportive comments on co-accessibility prediction evaluation, which is indeed more challenging and has quite promising results. We have also corrected the figure legend of figure 2, which mistakenly indicated that the conformation data arises from H1-ESC cells while it is from GM12878. Chromatin-chromatin interaction predictions are usually performed in a group of single cells. On complex samples, these groups might have too few cells, which might impact the predictions. We are unaware of chromosomal conformation data matching the rare kidney cells identified in the UVO data. To investigate the performance of scOpen on fewer cells, we have down-sampled GM12878 scATAC-seq data (originally 992 cells) to 50% (496 cells) and 25% (248 cells) and repeated the analysis in Fig. 2e, f. We observe only a small decrease in AUPR in all evaluated methods, i.e., the AUPR for scOpen for 100%, 50%, and 25% of data was 0.104, 0.104, and 0.095 (see Reply Letter Fig, 3). These results indicate that chromatin interactions prediction is still reliable with ~250 cells. These are now included in the manuscript (**see Results section and Supplementary Fig. 6c**).

Of note, we can evaluate the chromatin-chromatin predictions on complex data sets (hematopoiesis and t-cells) using silver standards. Cicero also generates gene activity scores based on the predicated links. So the intuition is that if the prediction of DNA interaction is better, the estimated gene activity scores should distinguish different cell types better. We observed that scOpen indeed improved the chromatin-chromatin interaction for all benchmarking datasets, including a complex sample (PBMC), as a higher clustering accuracy was obtained using the scOpen estimated matrix as input for Cicero comparing with the raw count matrix (**Fig. 2c-d; Supplementary Fig. 5**).

Reply Letter Fig.3. (a) Precision-recall curves showing the evaluation of the predicted links on GM12878 cells using the raw and imputed matrix as input after down-sampling to 50%. We used data from pol-II ChIA-PET (left) and Hi-C (right) as true labels. Colors refer to methods. We reported the AUPR for the top 3 methods. (b) Same as (a) for data down-sampled to 25%.

4. The authors compared scOpen to a wide variety of scATAC analysis tools, but did not include AtacWorks (Lal et al. PMID: 33686069). AtacWorks is a deep learning-based tool built on the Nvidia parabricks architecture that can denoise scATAC datasets to improve peak calling (especially in rare cell types). This is a different approach that the authors should consider commenting on in their discussion.

We appreciate this suggestion. AtacWorks is a supervised learning approach to denoise (pseudo) bulk ATAC-seq data, and it needs at least a high-quality ATAC-seq dataset as input for training. The outputs of AtacWorks are a signal track representing the enhanced ATAC-seq signal on the genome and the predicted peaks. Since scOpen and AtacWorks are very different in terms of the methodology, i.e., bulk vs. single cell, we didn't include it in our benchmarking. We mention this method now in the discussion.

5. NFkB signaling has been implicated in murine proximal tubule IRI and in human dedifferentiated PT (the latter based on multiomics, similar to this study). Did the authors detect any changes in NFkB TF activity in their UO time course?

We agree this is an interesting question, and indeed we detected changes in NFkB activity. The NFkB1 TF shows high activity in injured PT and lymphoid cells, indicating shared regulatory programs in these cell types (Fig. 3d-e). Moreover, it also shows a gradual increase over time in injured PT, suggesting that Nfkb1 signaling might have a dominating role in sustaining the injured PT phenotype. We have included the figures of Nfkb1 footprints profiles across cell types and time points (Fig. 3e-f) and revised the results section to reflect this (see text below).

“We also detect the high activity of Nfkb1 in injured PTs (and lymphocytes), which fits with the known role of Nfkb1 in injured/failed repair PTs. Moreover, our analysis also shows a gradual TF activity increase over time in injured PT (Fig. 3f), suggesting that Nfkb1 plays an important role in sustaining the injured PT phenotype.”

6. The authors state that they consider all genes with a TF footprint-supported Runx1 binding site as a Runx1 target. What proportion of these sites have a validated (eg. ChIP-seq or other

method) Runx1 binding site in a public database like ENCODE? The correlation between peak-to-gene following scOpen imputation was modest with $R=0.4$ (Fig 4J).

We thank the reviewer for the suggestion. We have checked the ENCODE database and found that the RUNX1 ChIP-seq data is only available for human K562 cells, which is quite distinct from our data (i.e., mouse kidney myofibroblasts). We are not aware of RUNX1 ChIP-seq on myofibroblasts or similar cells. On the other hand, we observe a significant association of genes up-regulated upon RUNX1 overexpression and predicted target genes (Fig. 4h), which supports our footprinting predictions and scOpen peak-to-gene links.

Did the authors observe any difference in correlation for peak-to-gene relationships depending on the location of the peak. relative to the TSS. For example, did peaks located within 50kb of the TSS correlate better with respect to peak-to-gene relationships? How about distal intergenic peaks vs. proximal peaks? Or peaks that overlap with known enhancer marks (eg. H3K27Ac)?

The association of link size and type of regulatory features is an important point. We observe only a marginal decrease in correlation over link size/distance (Reply Letter 3a). We also observe a non-significant trend of higher correlation in distal regions than promoters when using gene annotation to classify genomic regions as distal, exonic, intronic, or promoters (Reply Letter Fig. 4b). Finally, we used H3K27ac and H3K4me1 histone marks of mouse kidneys (ENCODE ENCSR000CDG & ENCSR436FYE) to defined active enhancers (H3K4me1 and H3K27ac) vs. non-enhancers. We observed that the associated active enhancers show a higher correlation than poised peaks (Reply Letter Fig. 4c). Altogether, these results indicate the low influence of the peak distance towards correlation but an increase in correlation for active enhancers. We have included these results in the manuscript (see results section and Supplementary Fig. 13).

Reply Letter Fig. 4. (a) Scatter plot showing the peak-to-gene correlation and the distance of ATAC-seq peaks to TSS. A line is fitted to show the pattern. (b) Boxplot showing the peak-to-gene correlation distribution. Peaks were annotated as distal, exonic, intronic, and promoter. (c) Same as (b). Peaks were classified as (active) enhancers if overlapping with H3k27Ac and H3K4me1 ChIP-seq peaks from mouse kidneys; otherwise, non-enhancers if only H3K27ac is present.

7. The Runx1 data is presented as if it is a new finding, but this TF is already known to regulate the myofibroblast transition (PMID 25313057) including from the authors own work

(BioRxiv411686). This is somewhat misleading, and this needs to be placed into context based on the prior published work.

We appreciate this question. It has been previously shown that Runx1 is involved in myofibroblast differentiation using human prostate-derived mesenchymal stem cells (MSC) and bone marrow MSC (PMID 25313057). Furthermore, it has been shown in zebrafish that Runx1 plays a role in scar deposition following cryo-injury (PMID 32341028). However, our data is the first to show this in an unbiased approach in mammals in vivo. We have performed the validation in a human kidney-derived myofibroblast cell line. Based on this comment, we have added the following paragraph to the discussion section:

“Runx1 has recently been reported as a potential inducer of EMT in proximal tubular cells (Zhou, T. et al. 2018). Furthermore, in-vitro data of mesenchymal stem cells~(MSCs) isolated from bone marrow or prostate gland points towards a potential myofibroblast differentiation role of Runx1 (Kim et al. 2014). In-vivo evidence for a functional role of Runx1 in regulating fibrogenesis has been demonstrated in zebrafish (Koth, et al. 2020). Single-cell RNA-seq data from zebrafish heart after cryo-injury suggests that endocardial cells and thrombocytes upregulate Runx1 while Runx1 mutant zebrafish demonstrated enhanced cardiac regeneration after cryoinjury with an ameliorated fibrotic response. Here we show for the first time in-vivo and in-vitro evidence that Runx1 in myofibroblasts regulates scar formation following a fibrogenic kidney injury in mice. Runx1 deficiency caused reduced myofibroblast formation and enhanced recovery”

REVIEWERS' COMMENTS

Reviewer #3 (Remarks to the Author):

The authors have addressed all my concerns, I congratulate them for their exhaustive work.

Reviewer #5 (Remarks to the Author):

The authors have very comprehensively addressed our concerns with a variety of new data and/ or adjustments to the text. Ben Humphreys with Parker Wilson.

Reply to Referees

We have implemented all editorial requests in this revised version including a better contextualization of scOpen with competing methods in the introduction, as well as an exact explanation of statistics used in all figures. We thank the editors for their detailed revision of the manuscript.

Reviewer #3 (Remarks to the Author):The authors have addressed all my concerns, I congratulate them for their exhaustive work.

Reviewer #5 (Remarks to the Author): The authors have very comprehensively addressed our concerns with a variety of new data and/ or adjustments to the text. Ben Humphreys with Parker Wilson.

We thank the referees for their constructive critics, which were essential to improve our manuscript.